# LEARNING TO PREDICT PARAMETER FOR UNSEEN DATA

## ABSTRACT

Typical deep learning models depend heavily on large amounts of training data and resort to an iterative optimization algorithm (e.g., SGD or Adam) for learning network parameters, which makes the training process very time- and resource-intensive. In this paper, we propose a new training paradigm and formulate network parameter training into a prediction task: given a network architecture, we observe there exists correlations between datasets and their corresponding optimal network parameters, and explore if we can learn a hyper-mapping between them to capture the relations, such that we can directly predict the parameters of the network for a new dataset never seen during the training phase. To do this, we put forward a new hypernetwork with the purpose of building a mapping between datasets and their corresponding network parameters, and then predict parameters for unseen data with only a single forward propagation of the hypernetwork. At its heart, our model benefits from a series of GRU sharing weights to capture the dependencies of parameters among different layers in the network. Extensive experimental studies are performed and experimental results validate our proposed method achieves surprisingly good efficacy. For instance, it takes 119 GPU seconds to train ResNet-18 using Adam from scratch and the network obtains a top-1 accuracy of 74.56%, while our method costs only 0.5 GPU seconds to predict the network parameters of ResNet-18 achieving comparable performance (73.33%), more than 200 times faster than the traditional training paradigm.

## 1 INTRODUCTION

Deep learning has yielded superior performance in a variety of fields in the past decade, such as computer vision (Kendall & Gal, 2017), natural language processing (DBL), reinforcement learning (Zheng et al., 2018; Fujimoto et al., 2018), etc. One of the keys to success for deep learning stems from huge amounts of training data used to learn a deep network. In order to optimize the network, the traditional training paradigm takes advantage of an iterative optimization algorithm (e.g., SGD) to train the model in a mini-batch manner, leading to huge time and resource consumption. For example, when training RestNet-101 (He et al., 2016) on the ImageNet (Deng et al., 2009) dataset, it often takes several days or weeks for the model to be well optimized with GPU involved. Thus, how to accelerate the training process of the network is an emergent topic in deep learning.

Nowadays, many methods for accelerating training of deep neural networks have been proposed (Kingma & Ba, 2015; Ioffe & Szegedy, 2015; Chen et al., 2018). The representative works include optimization based techniques by improving the stochastic gradient descent (Kingma & Ba, 2015; Yong et al., 2020; Anil et al., 2020), normalization based techniques (Ioffe & Szegedy, 2015; Salimans & Kingma, 2016; Ba et al., 2016), parallel training techniques (Chen et al., 2018; Kim et al., 2019), et. Although these methods have showed promising potential to speed up the training of the network, they still follow the traditional iterative-based training paradigm.

In this paper, we investigate a new training paradigm for deep neural networks. In contrast to previous works accelerating the training of the network, we formulate the parameter training problem into a prediction task: given a network architecture, we attempt to learn a hyper-mapping between datasets and their corresponding optimal network parameters, and then leverage the hyper-mapping to directly predict the network parameters for a new dataset unseen during training. A basic

assumption behind the above prediction task is that there exists correlations between datasets and their corresponding parameters of a given network. In order to demonstrate the rationality of this assumption, we perform the following experiment: for a dataset, we first randomly sample 3000 images to train a 3-layer convolutional neural network until convergence. Then we conduct the average pooling operation to the original inputs as a vector representation of the training data. We repeat the above experiment 1000 times, and thus obtain 1000 groups of representations and the corresponding network parameters. Finally, we utilize Canonical Correlation Analysis (CCA) (Weenink, 2003) to evaluate the correlations between training data and the network parameters by the above 1000 groups of data. Figure 1 shows the results, which illustrates there are indeed correlations between training datasets and their network parameters for a given network architecture.

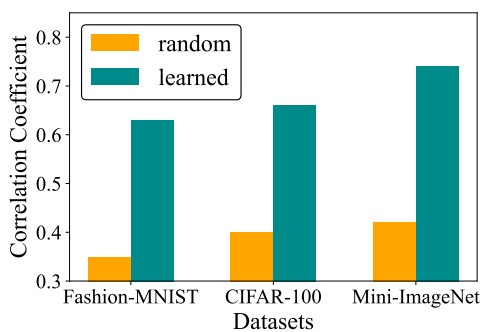

Figure 1: Correlation coefficients between training datasets and the network parameters on the Fashion-MNIST (Xiao et al., 2017), CIFAR-100 (Krizhevsky et al., 2009), Mini-ImageNet (Vinyals et al., 2016) datasets, respectively.'learned' depicts correlations between training datasets and the corresponding optimal network parameters. 'random' denotes correlations between training datasets and the network parameters selected randomly from 1000 groups.

In light of this, we propose a new hypernetwork, called PudNet, to learn a hyper-mapping between datasets and network parameters. Specifically, PudNet first summarizes the characters of datasets by compressing them into different vectors as their sketch. Then, PudNet extends the traditional hypernetwork (Ha et al., 2017) to predict network parameters of different layers based on these vectors. Considering that parameters among different layers should be dependent, we incorporate Gate Recurrent Unit (GRU) (Cho et al., 2014) into PudNet to capture the relations among them, so as to improve the performance of PudNet. Finally, it is worth noting that for training PudNet, it is infeasible if we prepare thousands of datasets and train networks on these datasets to obtain the corresponding optimal parameters respectively. Instead, we adopt a meta-learning based approach (Finn et al., 2017) to train the hypernetwork.

Extensive experiments demonstrate the surprising effectiveness of our PudNet. For example, it takes around 54, 119, 140 GPU seconds to train ResNet-18 using Adam from scratch and obtain top-1 accuracies of 99.91%, 74.56%, 71.84% on the Fashion-MNIST (Xiao et al., 2017), CIFAR-100 (Krizhevsky et al., 2009), Mini-ImageNet (Vinyals et al., 2016), respectively. While our method costs only around 0.5 GPU seconds to predict the parameters of ResNet-18 and still achieves 96.24%, 73.33%, 71.57% top-1 accuracies on the three datasets respectively, at least 100 times faster than the traditional training paradigm.

Our contributions are summarized as follows: 1) We find there are correlations between datasets and their corresponding parameters of a given network, and propose a general training paradigm for deep networks by formulating network training into a parameter prediction task. 2) We extend hypernetwork to learn the correlations between datasets and their corresponding network parameters, such that we can directly generate parameters for arbitrary unseen data with only a single forward propagation. 3) Our method achieves surprisingly good performance for unseen data, which is expected to motivate more researchers to explore along with this research direction.

## 2 RELATED WORK

### 2.1 HYPERNETWORKS

The original goal of hypernetwork proposed in (Ha et al., 2017) is to decrease the number of training parameters , by training a hypernetwork with a smaller size to generate the parameters of another network with a larger size on a fixed dataset. Because of its promising performance, hypernetwork has been gradually applied to various tasks (Krueger et al., 2017; Zhang et al., 2019; von Oswald et al., 2020; Li et al., 2020; Shamsian et al., 2021). von Oswald et al. (2020) proposes a task-conditioned hypernetwork to overcome catastrophic forgetting in continual learning. It learns an embedding for each task and utilizes the task embedding to generate corresponding parameters

for each task. Bayesian hypernetwork (Krueger et al., 2017) is proposed to approximate Bayesian inference in neural networks. GHN-2 proposed in Knyazev et al. (2021) attempts to build a mapping between the network architectures and network parameters, where the dataset is always fixed. GHN-2 leverages graph neural networks to model the information of the network architectures for learning the mapping. Our work is orthogonal to GHN-2, since we aim to build a mapping between the datasets and the network parameters, given a network architecture. Moreover, we extend the traditional hypernetwork by incorporating GRU to capture the relations among parameters of differnt layers and develop a meta-learning based manner to optimize the hypernetwork.

## 2.2 Acceleration of Network Training

Many works have been proposed to speed up the training process of deep neural networks in the past decade, including optimization based methods (Kingma & Ba, 2015; Yong et al., 2020; Anil et al., 2020), normalization based methods (Ioffe & Szegedy, 2015; Ba et al., 2016), parallel training methods (Chen et al., 2018; Kim et al., 2019), etc. Optimization based methods mainly aim to improve the stochastic gradient descent. For instance, Yong et al. (2020) proposes a gradient centralization method that centralizes gradient vectors to improve the Lipschitzness of the loss function. Normalization based methods intend to propose good normalization methods to speed up the training process. The representative work is the batch normalization (Ioffe & Szegedy, 2015) that can make the optimization landscape smooth and lead to fast convergence (Santurkar et al., 2018). Parallel training methods usually stack multiple hardwares to conduct parallel training, which can reduce training time by dispersing calculation amounts to distributed devices. However, these methods still follow the traditional iterative based training paradigm. Different from them, we attempt to explore a new training paradigm, and transform the network training problem of into a prediction task.

## 3 Proposed Method

In this section, we will introduce our PudNet in detail. For better illustration, we first give preliminaries and our problem formulation, and then elaborate the details of our method.

## 3.1 Preliminaries

**Notation** We denote $H_\theta$ as our hypernetwork parameterized by $\theta$. Let $\mathcal{D}^{train} = \{D_i\}_{i=1}^{\mathcal{N}}$ be the set of training datasets, where $D_i$ is the $i^{th}$ dataset and $\mathcal{N}$ is the number of training datasets. Each sample $x_j \in D_i$ has a label $y_j \in \mathcal{C}_i^{tr}$, where $\mathcal{C}_i^{tr}$ is the class set of $D_i$. We use $\mathcal{C}^{tr} = \bigcup_{i=1}^{\mathcal{N}} \mathcal{C}_i^{tr}$ to denote the whole label set of training datasets. Similarly, we define $\mathcal{D}^{test}$ as the set of unseen datasets used for testing and $\mathcal{C}^{te}$ as the set containing all labels in $\mathcal{D}^{test}$.

**Problem Formulation** In contrast to traditional iterative-based training paradigm, we attempt to explore a new training paradigm, and formulate the network training into a parameter prediction task. To this end, we propose the following objective function:

$$\arg\min_\theta \sum_{i=1}^{\mathcal{N}} \mathcal{L}(\mathcal{F}(D_i, \Omega; H_\theta), \mathcal{M}_i^\Omega), \tag{1}$$

where $\mathcal{F}(D_i, \Omega; H_\theta)$ denotes a forward propagation of our hypernetwork $H_\theta$. The input of the forward propagation is the dataset $D_i$ and its output is the predicted parameters of network $\Omega$ by $H_\theta$. Note that the architecture of $\Omega$ is always fixed during training and testing, e.g., ResNet-18. This makes sense because we often apply a representative deep model to data of different domains. Thus, it is obviously meaningful if we can predict the network parameters for unseen data using an identical network architecture. $\mathcal{M}^\Omega = \{\mathcal{M}_i^\Omega\}_{i=1}^{\mathcal{N}}$ denotes the ground-truth parameter set of network $\Omega$ corresponding to datasets $\mathcal{D}^{train}$, where $\mathcal{M}_i^\Omega$ is the ground-truth parameters for the dataset $D_i$. $\mathcal{L}$ is a loss function, measuring the difference between the ground-truth parameters $\mathcal{M}_i^\Omega$ and the predicted parameters.

The core idea in (1) is to learn a hyper-mapping $H_\theta$ between datasets $\mathcal{D}^{train}$ and the network parameter set $\mathcal{M}^\Omega$, on the basis of our finding that there are correlations between datasets and the network parameters, as shown in Figure 1. However, it is prohibitive if preparing thousands of datasets $D_i$

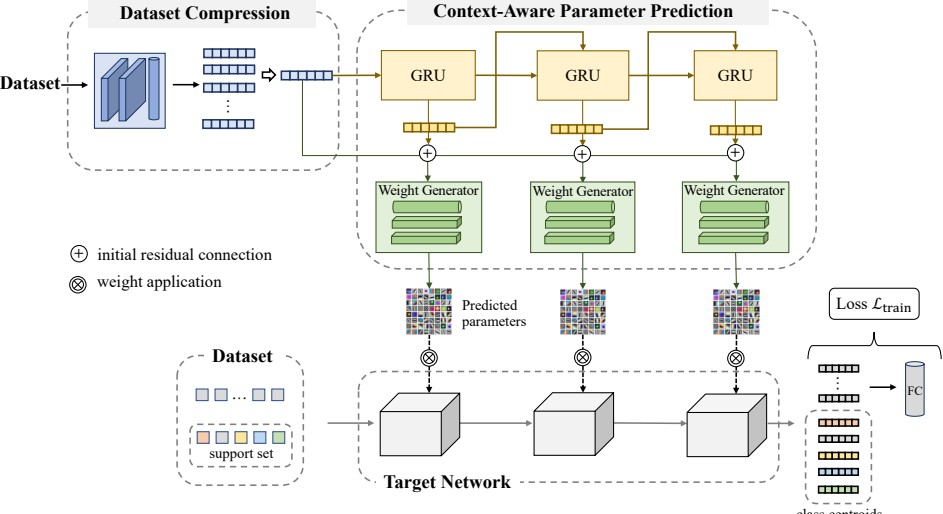

Figure 2: Overview of our proposed PudNet. PudNet first compresses each dataset into a sketch with a fixed size, and then utilizes the hypernetwork to generate parameters of a target network based on the sketch. Finally, PudNet is optimized based on a support set in a meta-learning based manner.

and training network $\Omega$ on $D_i$ to obtain the corresponding ground-truth parameters $\mathcal{M}_i^\Omega$ respectively. To alleviate this problem, we adopt a meta-learning based (Vinyals et al., 2016) approach to train the hypernetwork $H_\theta$, and propose another objective function as:

$$\arg\min_\theta \quad \sum_{i=1}^{\mathcal{N}} \sum_{x_j \in D_i} \mathcal{L}(x_j, y_j; \mathcal{F}(D_i, \Omega; H_\theta)), \tag{2}$$

Instead of optimizing $H_\theta$ by directly matching the predicted parameters $\mathcal{F}(D_i, \Omega; H_\theta)$ with the ground-truth parameters $\mathcal{M}_i^\Omega$, we can adopt a typical loss, e.g., cross-entropy, to optimize $H_\theta$, where each dataset $D_i$ can be regarded as a task in meta-learning (Vinyals et al., 2016). By learning on multiple tasks, the parameter predictor $H_\theta$ is gradually able to learn to predict performant parameters for training datasets $\mathcal{D}^{train}$. During testing, we can utilize $\mathcal{F}(D, \Omega; H_\theta)$ to directly predict the parameters for a testing dataset $D$ never seen in $\mathcal{D}^{train}$ with only a single forward propagation.

## 3.2 OVERVIEW OF OUR FRAMEWORK

Our goal is to learn a hypernetwork $H_\theta$, so as to directly predict the network parameters for an unseen dataset by $H_\theta$. However, there remains two issues that are not solved: First, the sizes of different $D_i$ may be different and the dataset sizes may be large, which makes $H_\theta$ hard to be trained; Second, there may be correlations among parameters of different layers in a network. However, how to capture such context relations among parameters has not been fully explored so far.

To this end, we propose a novel framework, PudNet, as shown in Figuire 2, PudNet first introduces a dataset compression module to compress each dataset $D_i$ into a small size sketch $s_i \in \mathbf{R}^{l \times m}$ to summarize the major characteristics of $D_i$, where $l$ and $m$ are the size and dimension of the sketch, respectively. Then, our context-aware parameter prediction module takes the sketch $s_i$ as input, and outputs the predicted parameters of the target network, e.g., ResNet-18. At its heart, multiple GRUs sharing weights are constructed to capture the dependencies of parameters among different layers in the network. Finally, PudNet is optimized based on a support set in a meta-learning based manner.

## 3.3 DATASET COMPRESSION

To solve the issue of different sizes of training datasets, we first compress each dataset into a sketch with a fixed size. In recent years, many data compression methods have been proposed, such as matrix sketching (Liberty, 2013; Qian et al., 2015), random projection (Sarlos, 2006; Liberty et al., 2007), etc. In principle, these methods can be applied to our data compression module. For simplification, we leverage a deep neural network to extract a feature vector as the representation of each

sample, and then conduct the average pooling operation to generate a final vector as the sketch of the dataset. The sketch $s_i$ for the dataset $D_i$ can be calculated as:

$$s_i = \frac{1}{|D_i|} \sum_{x_j \in D_i} T_\phi(x_j),$$

(3)

where $T_\phi(\cdot)$ denotes a feature extractor parameterized by $\phi$, and the structure of the feature extractor used in the experiments can be found in Appendix A.1. $|D_i|$ is the size of the dataset $D_i$. The parameter $\phi$ is jointly trained with PudNet in an end-to-end fashion. In future work, more efforts could be made to explore more effective solutions to summarize the information of a dataset, e.g. using statistic network (Edwards & Storkey, 2017) to compress datasets.

### 3.4 CONTEXT-AWARE PARAMETER PREDICTION

After obtaining the sketches for all training datasets, we will feed them into the context-aware parameter prediction module, i.e., our hypernetwork, as shown in Figure 2. In the followings, we will introduce this module in detail.

**Capturing Context Relations via GRU**  Since an input of a neural network would sequentially pass forward the layers of this network, the parameters of different layers should be not independent. If we ignore the context relations among parameters of different layers, the solution may be suboptimal. Thus, we utilize GRU (Cho et al., 2014) to capture the context-aware parameter relations as shown in Figure 3. Note that we have two changes compared to conventional GRU: 1) conventional GRU utilizes the randomly initialized hidden state $h_0$. Different from this, we set the dataset sketch embedding as the initial hidden state: $h_0 = s_i$, so as to provide dataset information for predicting corresponding structure parameters; 2) In each recurrent step, conventional GRU usually takes the next word as input in natural language process field (Cho et al., 2014). Instead of that, we exploit the predicted structure parameters of previous layer as input, enabling the information of shallower layer parameters memorized in GRU to help the prediction of parameters in the deeper layer. By this way, the dependency relations among parameters of different layers can be well captured.

The following comes a formal description:

$$r_t = \sigma(W_r \cdot [h_{t-1}, a_{t-1}]),$$
$$z_t = \sigma(W_z \cdot [h_{t-1}, a_{t-1}]),$$
$$\tilde{h}_t = tanh(W_h \cdot [r_t * h_{t-1}, a_{t-1}]),$$
$$h_t = (1 - z_t) * h_{t-1} + z_t * \tilde{h}_t,$$
$$a_t = \sigma(W_o \cdot h_t),$$

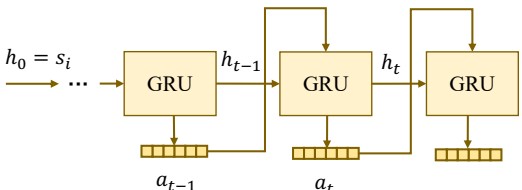

Figure 3: An illustration of capturing context relations via GRU.

where $h_t$ could transfer dataset related information. $a_t$ is a latent vector encoding the context information. $W_r, W_z, W_h, W_o$ are the learnable parameters. As in GRU, the reset gate $r_t$ decides how much information in the hidden state $h_{t-1}$ needs to be reset. $\tilde{h}_t$ is a new memory, which absorbs the information of $h_{t-1}$ and $a_{t-1}$. $z_t$ is an update gate, which regulates how much information in $\tilde{h}_t$ to update and how much information in $h_{t-1}$ to forget. The context-aware output $a_t$ is taken as input to the weight generator to predict parameters of the $t$-th layer of the target network $\Omega$.

**Initial Residual Connection**  To ensure that the final context-aware output contains at least a fraction of the initial dataset information, we additionally implement an initial residual connection between dataset sketch embedding $s_i$ and $a_t$ as:

$$\hat{a}_t = a_t \times (1 - \eta) + s_i \times \eta,$$

(4)

where $\eta$ is the hyperparameter. After obtaining $\hat{a}_t$, we put $\hat{a}_t$ into a weight generator used to generate the parameters of $\Omega$.

**Weight Generator**  Since the target network $\Omega$ usually has different sizes and dimensions in different layers, we construct the weight generator $g_{\psi_t}$ for each layer $t$ to transform $\hat{a}_t$ of a fixed dimension to network parameter tensor $w_t$ with variable dimensions. Here $g_{\psi_t}$ denotes the weight

generator of the $t$-th layer, $\psi_t$ is the learnable parameters of $g_{\psi_t}$, $\boldsymbol{w}_t$ is the predicted parameter of $t$-th layer in $\Omega$. We can derive the predicted parameter of the $t$-th layer as:

$$\boldsymbol{w}_t = g_{\psi_t}(\hat{\boldsymbol{a}}_t) \tag{5}$$

where $g_{\psi_t}$ consists of one linear layer and two $1 \times 1$ convolutional layers. More details of the weight generator can be found in Appendix A.1. When the parameters $\boldsymbol{w}_t$ of each layer are predicted, we can use these parameters as the final parameters of $\Omega$.

## 3.5 OPTIMIZATION OF OUR FRAMEWORK

In this section, we introduce how to optimize our PudNet. In contrast to traditional classification tasks where training data and testing data have the identical label space. In our task, the label spaces between training and testing can be different, even not overlapped. Thus, training a classification head on the training data can not be used to predict labels of testing data. Motivated by several metric learning methods (Chen et al., 2021; Oreshkin et al., 2018), we introduce a parameter-free classification method to solve the above issue.

Similar to Chen et al. (2021), we obtain a metric-based category prediction on class $c_k$ as:

$$p(y = c_k | x_j, \Omega, D_i; H_\theta) = \frac{exp(\tau \cdot < f(x_j; \mathcal{F}(D_i, \Omega; H_\theta)), \boldsymbol{u}_k >)}{\sum_c exp(\tau \cdot < f(x_j; \mathcal{F}(D_i, \Omega; H_\theta)), \boldsymbol{u}_c >)}, \tag{6}$$

where $\boldsymbol{u}_k$ is the centroid of class $c_k$, which is the average output of the predicted network $\Omega$ over samples belonging to class $c_k$ in the support set, as in Snell et al. (2017). $< \cdot, \cdot >$ denotes the cosine similarity of two vector, and $\tau$ is a learnable temperature parameter. $f(x_j; \mathcal{F}(D_i, \Omega; H_\theta))$ is the output of the target neural network $\Omega$ based on the input $x_j$.

Then, the parameter-free classification loss can be defined as:

$$\mathcal{L}_1 = \sum_{i=1}^{\mathcal{N}} \sum_{x_j \in D_i} \mathcal{L}(p(y|x_j, \Omega, D_i; H_\theta), y_j), \tag{7}$$

where $y_j$ is the true label of $x_j$, $\mathcal{L}$ is the cross-entropy loss.

To further improve the performance of the model, we introduce an auxiliary task for training our hypernetwork, by adding a full classification head $\mathcal{Q}_\varphi$ parameterized by $\varphi$. The classification head aims to map the output of the target network $\Omega$ to probabilities of the whole classes $\mathcal{C}^{tr}$ from $\mathcal{D}^{train}$. The full classification loss is defined as:

$$\mathcal{L}_2 = \sum_{i=1}^{\mathcal{N}} \sum_{x_j \in D_i} \mathcal{L}(\mathcal{Q}_\varphi(f(x_j; \mathcal{F}(D_i, \Omega; H_\theta))), y_j). \tag{8}$$

Our parameter prediction task co-trained with a full classification head is related to curriculum learning (Oreshkin et al., 2018). Since learning on varying label space is more challenging than learning on a static one, the full classification problem that maps features to a static label set could be regraded as a simpler curriculum. This easier 'prerequisite' could help the hypernetwork to obtain the basic level knowledge before handling harder parameter prediction task. Moreover, to make parameter-free based prediction and full classification based prediction consistent, which is motivated by (Chen et al., 2022; Wu et al., 2019), we introduce a Kullback-Leibler Divergence loss to encourage their predicted probabilities to be matched:

$$\mathcal{L}_3 = \sum_{i=1}^{\mathcal{N}} \sum_{x_j \in D_i} KL(q(y|x_j)||p(y|x_j)), \tag{9}$$

where $KL$ is the Kullback-Leibler Divergence. $p(y|x_j)$ and $q(y|x_j)$ are the predicted probabilities of $x_j$ of parameter-free based and full classification based methods, respectively. The probabilities of the corresponding classes in $p(y|x_j)$ are padding with zero to match the dimension of $q(y|x_j)$.

Finally, we give the overall multi-task loss as:

$$\mathcal{L}_{total} = \mathcal{L}_1 + \mathcal{L}_2 + \mathcal{L}_3. \tag{10}$$

where $D_i$ can be regarded as a task similar to that in meta-learning. By minimizing (10), our hypernetwork can be well trained. For an unseen data in testing, we utilize our hypernetwork to directly predict its network parameters, and use the parameter-free based method for classification. The training procedure of our PudNet is in Appendix A.4.

# 4 EXPERIMENT

## 4.1 DATASET CONSTRUCTION

In the experiment, we construct numerous datasets for evaluating our method based on four datasets: Fashion-MNIST (Xiao et al., 2017), CIFAR-100 (Krizhevsky et al., 2009), Mini-ImageNet (Vinyals et al., 2016), Animals-10 (Gupta & Brown, 2022). The constructed datasets are summarized as:

**Fashion-set**: We randomly select 6 classes from Fashion-MNIST to construct training datasets and the remaining 4 classes for constructing testing datasets. We construct 2000 groups of datasets from the 6-category training set to train PudNet. To verify PudNet's ability to directly generate parameters, we construct 500 groups of datastes from the 4-category testing set to generate 500 groups of network parameters. For each group of network parameters, we also construct another dataset having identical labels but not overlapped samples with the dataset used for generating parameters, in order to test the performance of the predicted network parameters. Each dataset contains 600 randomly sampled images with 2 randomly sampled classes.

**CIFAR100-set**: We randomly choose 80 classes from CIFAR-100 for constructing training datasets and 20 classes not overlapped with the above 80 classes to construct testing datasets. We sample 100000 groups of datasets for training PudNet. Similar to Fashion-set, we construct 500 groups of datastes to directly generate their network parameters by PudNet, and create another 500 groups of datasets for testing the performance of the predicted parameters. Each datasets consists of 500 images with 5 classes randomly selected.

**ImageNet-set**: Similar to CIFAR100-set, the mini-Imagenet dataset is randomly split into 80 classes for creating training datasets and 20 classes for creating testing datsets. We sample 50000 groups of datasets for training PudNet. Similar to Fashion-set, we use 500 groups of datastes to generate their network parameters, and construct another 500 groups of datasets for testing. There are 500 images with 5 classes selected randomly in each dataset.

**CIFAR100→Animals10**: To further verify our PudNet, we construct a cross-domain dataset. We use CIFAR100-set to construct training datasets, and Animals-10 for testing datasets. There are 100000 groups of datasets from CIFAR100-set to train PudNet. We randomly split Animals10 into two not overlapped subsets: one is used to generate parameters, and the other for testing.

## 4.2 BASELINES

We compare our method with traditional iterative based training paradigm including training from scratch and one training acceleration method, GC (Yong et al., 2020). We also take the pretrained model as a baseline. In addition, we also compare with meta-learning methods, including Match-Net (Vinyals et al., 2016), ProtoNet (Snell et al., 2017), Meta-Baseline (Chen et al., 2021), Meta-DeepDBC (Xie et al., 2022), and MUSML (Jiang et al., 2022). We use two kinds of architectures as our target network $\Omega$: a 3-layer CNN, ConvNet-3 and Resnet-18. To ensure a fair comparison, we use all training datasets for training meta-learning methods and the pretrained model. For all experiments, we use ACC (Top-1 Accuracy) metric to evaluate the classification performance.

## 4.3 IMPLEMENTATION DETAILS

We perform the experiments using GeForce RTX 3090 Ti GPU. We set the learning rate as 0.001. For the target network ConvNet-3, we set the hyperparameter $\eta$ as 0.2,0.1,0.3,0.1 for Fashion-set, CIFAR100-set, ImageNet-set, CIFAR100→Animals10 respectively. For the target network ResNet-18, we set the hyperparameter $\eta$ as 0.2,0.5,0.5,0.5 for Fashion-set, CIFAR100-set, ImageNet-set, CIFAR100→Animals10 respectively. More details could be found in Appendix A.1 and A.2.

## 4.4 RESULT AND ANALYSIS

**General Performance Analysis** Table 1 and Table 2 show the general results of our method. We could find that our method consistently outperforms the meta-learning methods and the pretrained method. This demonstrates that learning a hyper-mapping between datasets and corresponding network parameters is effective. To demonstrate the time consumption our method could save, we also

Table 1: Results of different methods in terms of the target network ConvNet-3 on the Fashion-set, CIFAR100-set, ImageNet-set datasets.

| Method | | Fashion-set | | CIFAR100-set | | ImageNet-set | |
|---|---|---|---|---|---|---|---|
| | | ACC(%) | time (sec.) | ACC | time (sec.) | ACC | time (sec.) |
| Pretrained | | 94.12±0.63 | - | 58.35±0.61 | - | 53.28±0.67 | - |
| MatchNet | | 89.93±0.65 | - | 47.75±0.73 | - | 43.83±0.83 | - |
| ProtoNet | | 92.32±0.37 | - | 51.96±0.57 | - | 49.59±0.88 | - |
| Meta-Baseline | | 94.85±0.31 | - | 57.69±0.38 | - | 54.97±0.75 | - |
| Meta-DeepDBC | | 95.76±0.39 | - | 60.52±0.41 | - | 55.36±0.73 | - |
| MUSML | | 96.05±0.32 | 1.21 | 56.49±0.56 | 1.22 | 54.03±0.94 | 1.22 |
| Adam Scratch | 1 epochs | 91.07±1.11 | 0.87 | 49.11±1.03 | 0.99 | 41.53±1.20 | 2.67 |
| | 30 epochs | 99.97±0.02 | 25.99 | 64.54±0.40 | 28.37 | 64.33±0.83 | 77.08 |
| | 50 epochs | 99.96±0.03 | 43.36 | 70.68±0.53 | 49.02 | 67.25±0.69 | 133.48 |
| GC | 1 epochs | 92.36±1.20 | 0.88 | 50.23±1.23 | 0.99 | 40.74±1.32 | 2.67 |
| | 30 epochs | 99.98±0.01 | 26.01 | 66.76±0.54 | 29.83 | 65.45±0.89 | 77.12 |
| | 50 epochs | 99.97±0.02 | 43.42 | 71.56±0.63 | 50.22 | 69.44±0.77 | 133.69 |
| PudNet | | 96.64±0.34 | 0.03 | 64.09±0.40 | 0.03 | 59.31±0.64 | 0.03 |

Table 2: Results of different methods in terms of the target network ResNet-18 on the Fashion-set, CIFAR100-set, ImageNet-set datasets.

| Method | | Fashion-set | | CIFAR100-set | | ImageNet-set | |
|---|---|---|---|---|---|---|---|
| | | ACC(%) | time (sec.) | ACC | time (sec.) | ACC | time (sec.) |
| Pretrained | | 93.76±0.47 | - | 64.58±0.59 | - | 65.67±0.73 | - |
| MatchNet | | 90.16±0.53 | - | 56.23±0.71 | - | 53.17±0.91 | - |
| ProtoNet | | 93.64±0.47 | - | 60.29±0.59 | - | 58.95±0.83 | - |
| Meta-Baseline | | 95.35±0.29 | - | 67.51±0.55 | - | 67.16±0.70 | - |
| Meta-DeepDBC | | 94.28±0.31 | - | 69.54±0.49 | - | 68.48±0.60 | - |
| MUSML | | 95.87±0.44 | 2.55 | 66.47±0.63 | 2.59 | 66.03±0.91 | 2.60 |
| Adam Scratch | 1 epochs | 93.98±1.21 | 1.83 | 52.82±1.01 | 3.96 | 46.43±1.18 | 4.81 |
| | 30 epochs | 99.91±0.05 | 54.22 | 74.56±0.45 | 118.87 | 71.84±0.69 | 140.37 |
| | 50 epochs | 99.87±0.11 | 91.19 | 79.85±0.47 | 198.12 | 75.98±0.71 | 231.63 |
| GC | 1 epochs | 94.11±1.25 | 1.88 | 53.21±1.23 | 4.01 | 47.55±1.33 | 4.82 |
| | 30 epochs | 99.94±0.05 | 54.93 | 75.74±0.59 | 119.03 | 72.89±0.73 | 140.98 |
| | 50 epochs | 99.96±0.03 | 91.73 | 79.98±0.55 | 199.61 | 76.73±0.87 | 232.57 |
| PudNet | | 96.24±0.39 | 0.50 | 73.33±0.54 | 0.49 | 71.57±0.71 | 0.50 |

Table 3: Results of different methods on the cross-domain datasets CIFAR100→Animals10

| Method | | ConvNet-3 | | ResNet-18 | |
|---|---|---|---|---|---|
| | | ACC (%) | time (sec.) | ACC (%) | time (sec.) |
| Pretrained | | 16.43±0.73 | - | 33.36±0.75 | - |
| Meta-DeepDBC | | 31.93±0.81 | - | 40.50±0.64 | - |
| MUSML | | 26.77±0.67 | 1.71 | 36.78±0.59 | 3.21 |
| Adam Scratch | 1 epochs | 18.33±0.57 | 21.37 | 22.09±0.58 | 31.29 |
| | 5 epochs | 38.71±0.31 | 103.25 | 49.12±0.08 | 156.56 |
| | 10 epochs | 53.62±0.18 | 203.64 | 66.44±0.37 | 311.74 |
| GC Scratch | 1 epochs | 18.94±0.64 | 21.33 | 23.01±1.02 | 30.79 |
| | 5 epochs | 39.43±0.44 | 102.73 | 49.77±0.54 | 155.34 |
| | 10 epochs | 55.21±0.51 | 202.67 | 68.56±0.39 | 310.29 |
| PudNet | | 35.15±0.77 | 0.03 | 43.21±0.69 | 0.49 |

provide the time of training the model from scratch by a widely-used optimizer Adam (Kingma & Ba, 2015) and the training acceleration technique, GC. We could find that it takes around 55, 119, 140 GPU seconds to train ResNet-18 using the accelerated method GC and the network obtains top-1 accuracies of 99.94%, 75.74%, 72.89% on the Fashion-set, CIFAR-set, ImageNet-set respectively. While our method costs only around 0.5 GPU seconds to predict the parameters of ResNet-18 and still achieves a comparable performance (96.24%, 73.33%, 71.57% top-1 accuracies) on the three datasets respectively, at least 100 times faster than the accelerated method.

**Performance on Cross-domain Datasets** We further evaluate our method on the cross-domain datasets, CIFAR100→Animals10. Table 3 shows the results on the CIFAR100→Animals10 dataset.

Table 4: Ablation study of our method.

| Setting | ConvNet-3 | | | ResNet-18 | | |
|---|---|---|---|---|---|---|
| | Fashion-set | CIFAR100-set | ImageNet-set | Fashion-set | CIFAR100-set | ImageNet-set |
| PudNet-w.o.-Context | 93.13±0.36 | 59.51±0.42 | 54.36±0.67 | 93.08±0.44 | 65.35±0.51 | 61.42±0.70 |
| PudNet-Random | 90.51±0.32 | 57.26±0.39 | 52.88±0.59 | 89.09±0.40 | 60.56±0.50 | 55.93±0.70 |
| PudNet-w.o.-initRes | 95.46±0.35 | 61.56±0.40 | 57.81±0.66 | 95.42±0.44 | 71.06±0.51 | 68.98±0.73 |
| PudNet-metric | 95.31±0.45 | 60.97±0.53 | 50.15±0.87 | 94.75±0.44 | 68.60±0.61 | 61.28±0.85 |
| PudNet-w.o.-kl | 95.92±0.34 | 61.29±0.41 | 57.64±0.62 | 95.44±0.38 | 70.27±0.54 | 67.53±0.73 |
| PudNet | **96.64**±0.34 | **64.09**±0.40 | **59.31**±0.64 | **96.24**±0.39 | **73.33**±0.54 | **71.57**±0.71 |

we can obtain comparable accuracy when training the model from scratch at around 5 epochs, while our method is at least 300 times faster than the traditional training methods. In addition, we also compare with Meta-DeepDBC that generally achieves better performance based on Table 1 and 2 and with MUSML which is proposed very recently. Our model still outperforms them in a large margin. We expect that such a result could motivate more researchers to explore along this direction.

**Ablation Study**   We design a variant of our method to analyse the effect of the context relation information. PudNet-w.o.-Context denotes our method directly feeds the dataset sketch into the weight generator without using GRU. The results are listed in Table 4. Our PudNet outperforms PudNet-w.o.-Context in a large margin, demonstrating the effectiveness of capturing dependencies among parameters of different layers.

Moreover, we design two variants of our method to further study the contribution of the dataset information. PudNet-Random denotes our method randomly initializes the hidden state of GRU. PudNet-w.o.-initRes denotes our method does not utilize initial residual connection for initial dataset information complementary. As shown in Table 4, the performance of our method decreases when randomly initializing the hidden state of GRU. It indicates that exploiting the initialization of the hidden state to deliver the information of datasets is effective. Moreover, PudNet is better than PudNet-w.o.-initRes, illustrating the effectiveness of the initial residual connection.

Finally, we design another two variants of our methods to study the impact of the auxiliary task. PudNet-metric means our method only using the parameter-free loss. PudNet-w.o.-kl means out method without using the KL Divergence. As shown in Table 4, PudNet-w.o.-kl has better performance than PudNet-metric, demonstrating it is effective for the auxiliary full classification task. PudNet outperforms PudNet-w.o.-kl. This illustrates it is effective to encourage the predicted probability distribution of parameter-free method and full classification method to be matched.

**Fine-tuning Predicted Parameters**   Since a typical strategy for applying a pretrained model to a new dataset is to fine-tune the model. Thus, we intend to evaluate the performance of fine-tuning our method and baselines on the CIFAR-set dataset. To do this, we first incorporate an additional linear classification layer to our method and baselines, except 'From scratch'. Then, we randomly select 10000 samples from 20 classes for fine-tuning the models, and use the remaining 2000 samples for testing. Table 5 shows the results. Here 'From Scratch' means directly training the target network with random initialized parameters from scratch. Our method achieves the best performance. This indicates the predicted parameters by our PudNet can well serve as a pretrained model.

Table 5: Performance of finetuning all methods using 50 epochs.

| Method | ConvNet-3 | ResNet-18 |
|---|---|---|
| From Scratch | 44.85±0.23 | 48.25±0.30 |
| Pretrained | 49.40±0.17 | 59.75±0.24 |
| Meta-DeepDBC | 50.05±0.21 | 61.35±0.28 |
| PudNet | **55.21**±0.19 | **65.19**±0.22 |

## 5 CONCLUSION

In this paper, we found there are correlations among datasets and the corresponding parameters of a given network, and explored a new training paradigm for deep neural networks. We proposed a new hypernetwork, PudNet, which can directly predict the network parameters for an unseen data with only a single forward propagation. Essential to our hypernetwork is the construction of a series of GRU, to capture the relations among parameters of different layers in a network. Extensive experimental results demonstrated the effectiveness and efficiency of our method.

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

# A   APPENDIX

## A.1   ARCHITECTURE DETAILS

**Dataset Compression Module Details.** The feature extractor in our framework contains several basic blocks, where each basic block consists of a $5 \times 5$ convolutional layer, a leakyReLU function and a batch normalization layer. For generating parameters of ConvNet-3, we use one basic block as the feature extractor. For ResNet-18 or ResNet-34, we stack five basic blocks as the feature extractor. Note that our feature extractor is jointly trained with PudNet in an end-to-end manner.

**Structure of the Weight Generator**   Figure 4 shows the architecture of the weight generator. The weight generator takes as input a vector with the dimension of $d_a$, and outputs a tensor with the size of $d_{out} \times d_{in} \times f \times f$ as the parameters of the convolutional layer.

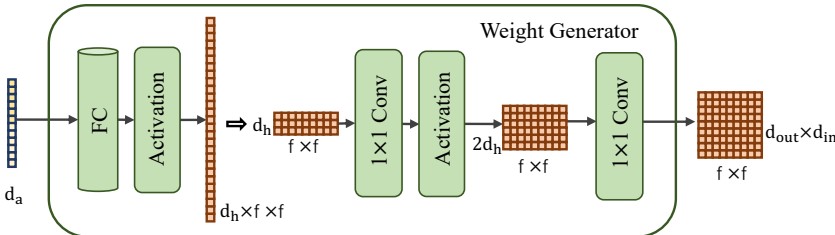

Figure 4: Architecture of the Weight Generator.

**Structures of the Target Networks**. We use ConvNet-3, ResNet-18, ResNet-34 as the target networks. The structure of ConvNet-3 is shown in Figure 5. For ResNet-18 and ResNet-34, we use the same architectures with He et al. (2016).

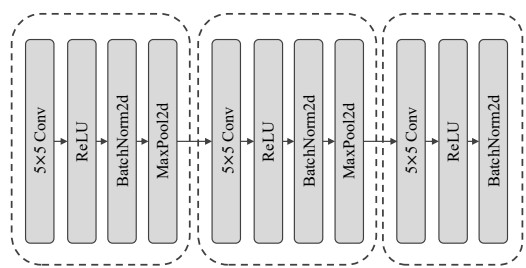

Figure 5: Structure of the Target Network ConvNet-3.

## A.2   ADDITIONAL IMPLEMENTATION DETAILS

For all experiments, we use ACC (Top-1 Accuracy) metric to evaluate the classification performance. All experiments are optimized by Adam optimizer. We set learning rate to 0.001 and train PudNet until convergence. In metric-based learning process, following Chen et al. (2021), the temperature $\tau = 10$ in Eq.( 6) is fixed. 10 labeled samples per class are used as support sets to deduce the class centroid. As mentioned in the main paper, we introduce auxiliary tasks to assist optimization. We add an full classification linear layer (e.g. 80-way linear head in CIFAR-100) to maintain static class set during training. We also introduce a consistency loss, while the dimension between logits deduced by metric-based classification (e.g.5-dimensional in CIFAR-100) and the logits produced by full linear head (e.g.80-dimensional in CIFAR-100) are not matching, thus we transpose the 5-dimensional logit to 80-dimensional logit, by padding the rest values with zero. We search $\eta$ from $\{0, 0.1, 0.2, 0.3, \cdots, 0.9\}$. For target network ConvNet-3, we set $\eta = 0.2$ for Fashion-set, $\eta = 0.1$ for CIFAR100-set, $\eta = 0.3$ for ImageNet-set, $\eta = 0.1$ for CIFAR100→Animals10. For target network ResNet-18, we set $\eta = 0.2$ for Fashion-set, $\eta = 0.5$ for CIFAR100-set and ImageNet-set, $\eta = 0.5$ for CIFAR100→Animals10.

Table 6: Results of different methods on new dataset CIFAR-100→CIFAR-10.

| Method | | ConvNet-3 | | ResNet-18 | |
|---|---|---|---|---|---|
| | | ACC(%) | time (sec.) | ACC (%) | time (sec.) |
| Pretrained | | 34.32±0.61 | - | 40.93±0.48 | - |
| Meta-DeepDBC | | 41.28±0.59 | - | 47.15±0.63 | - |
| MUSML | | 37.45±0.53 | 2.37 | 45.37±0.57 | 5.34 |
| Adam Scratch | 1 epochs | 19.98±0.63 | 51.57 | 21.37±0.71 | 60.79 |
| | 5 epochs | 44.67±0.42 | 253.16 | 33.76±0.44 | 310.29 |
| | 10 epochs | 48.69±0.48 | 506.63 | 48.21±0.25 | 621.75 |
| | 20 epochs | 58.34±0.53 | 1015.26 | 65.17±0.55 | 1248.23 |
| GC Scratch | 1 epochs | 20.03±0.83 | 51.74 | 21.44±1.02 | 60.99 |
| | 5 epochs | 44.89±0.67 | 253.56 | 34.41±0.78 | 310.33 |
| | 10 epochs | 49.36±0.55 | 507.12 | 49.89±0.53 | 622.01 |
| | 20 epochs | 58.74±0.49 | 1015.87 | 66.78±0.47 | 1248.98 |
| PudNet | | **43.76**±0.64 | 0.05 | **51.05**±0.56 | 0.56 |

Table 7: Results of different methods on the cross-domain datasets on ResNet-18.

| Method | | ImageNet→Animals10 | | ImageNet→CIFAR10 | |
|---|---|---|---|---|---|
| | | ACC (%) | time (sec.) | ACC (%) | time (sec.) |
| Pretrained | | 34.79±0.49 | - | 34.54±0.63 | - |
| Meta-DeepDBC | | 38.57±0.55 | - | 40.93±0.71 | - |
| MUSML | | 36.89±0.44 | 3.21 | 37.12±0.65. | 5.34 |
| Adam Scratch | 1 epochs | 22.09±0.58 | 31.29 | 21.37±0.71 | 60.79 |
| | 5 epochs | 49.12±0.08 | 156.56 | 33.76±0.44 | 310.79 |
| | 10 epochs | 66.44±0.37 | 311.74 | 48.21±0.25 | 621.75 |
| | 20 epochs | 73.47±0.67 | 623.92 | 65.17±0.55 | 1248.23 |
| GC Scratch | 1 epochs | 23.01±1.02 | 30.79 | 21.44±1.02 | 60.99 |
| | 5 epochs | 49.77±0.54 | 155.34 | 34.41±0.78 | 310.33 |
| | 10 epochs | 68.56±0.39 | 310.29 | 49.89±0.53 | 622.01 |
| | 20 epochs | 75.04±0.61 | 623.33 | 66.78±0.47 | 1248.98 |
| PudNet | | **42.43**±0.58 | 0.48 | **45.07**±0.70 | 0.57 |

### A.3 ADDITIONAL EXPERIMENTS

**Performance on Other Full Dataset Analysis:** Here we utilze CIFAR100-set to train the model and evaluate the performance on CIFAR-10. Note that the classes in CIFAR-10 are mutually exclusive with the classes in CIFAR-100(Krizhevsky et al., 2009). Since Meta-DeepDBC achieves better performance among all meta-learning methods based on Table 1 and 2, here we only report the results of Meta-DeepDBC and pretrained model for clarity. Table 6 shows the results on CIFAR-100→CIFAR-10. We observe that our model outperforms the Pretrained and Meta-DeepDBC competitors. This further verifies the effectiveness of our method. We also provide the time consumption of training the target network from scratch with Adam optimizer on CIFAR-10. It worth noting that it takes around 622 GPU seconds to train ResNet-18 using Adam from scratch and obtain top-1 accuracy of 48.21% on CIFAR-10. While our method costs only around 0.56 GPU seconds to predict the parameters of ResNet-18 and still achieves 51.05% top-1 accuracy, which is 1000 times faster than the training from scratch method.

**Performance on Cross-domain Datasets:** We construct another two cross-domain datasets. We use ImageNet-set to construct training datasets, and Animals-10 and CIFAR10 for testing datasets. There are 50000 groups of datasets from ImageNet-set to train PudNet. We randomly split CIFAR10 into two nonoverlapping subsets separately: one is used to generate parameters, and the other for testing. The separation process for Animals10 is analogy. We further evaluate our method on these two cross-domain datasets: ImageNet→Animals10, ImageNet→CIFAR10. Table 7 shows the results. our method still achieves surprisingly good efficiency. For instance, it takes 622.01 GPU seconds to train ResNet-18 on the ImageNet→CIFAR10 dataset using GC from scratch and obtain

a top-1 accuracy of 49.89%, while our method PudNet costs only 0.57 GPU seconds to predict the network parameters of ResNet-18 achieving comparable performance (45.07%), more than 1000 times faster than the traditional training paradigm. In addition, we could find that our method also outperforms state-of-the-art meta-learning methods in a large margin. We expect that such a result could motivate more researchers to explore along this direction.

**Performance on Deeper Target Network:** We perform another experiment to directly predict parameters of ResNet-34 by PudNet on CIFAR100-set. The results are listed in Table 8. We observe that our method achieves comparable performance to that of GC at 30 epochs, while our method is more than 250 times faster than GC. This further demonstrates the efficiency of our method.

Table 8: Results of different methods in terms of the target network ResNet-34 on CIFAR100-set.

| Method | | ACC(%) | time(sec.) |
|---|---|---|---|
| Pretrained | | 65.03±0.53 | - |
| Meta-baseline | | 67.40±0.69 | - |
| Meta-DeepDBC | | 69.64±0.75 | - |
| MUSML | | 66.39±0.59 | 3.11 |
| Adam Scratch | 1 epochs | 47.39±1.36 | 5.47 |
| | 30 epochs | 71.17±0.53 | 153.87 |
| | 50 epochs | 78.72±0.71 | 263.25 |
| GC Scratch | 1 epochs | 48.44±1.41 | 5.52 |
| | 30 epochs | 72.37±0.75 | 154.19 |
| | 50 epochs | 79.85±0.83 | 264.03 |
| PudNet | | 72.87±0.64 | 0.59 |

**Effect of Different Dataset Embedding:** To further study the effect of dataset embedding, we design three variants to predict parameters for ConvNet-3 on CIFAR100. "Sum" denotes summing up the representations of all samples in a dataset as the dataset embedding. "Geometric means" denotes using the geometric mean of sample representations as the dataset embedding. "Mean+Var" denotes concatenating the mean and the variance of sample representations as the dataset embedding. The results are reported in Table 9. We find that these four dataset embedding methods have comparable results. Here we only explore some simple dataset embedding methods. In the future, more complicated data compression methods could be explored, such as matrix sketching (Liberty, 2013; Qian et al., 2015), random projection (Sarlos, 2006; Liberty et al., 2007) and statistic network(Edwards & Storkey, 2017) , etc.

Table 9: Effect of Different Dataset Embedding

| Method | Sum | Geometric mean | Mean+Var | Mean(Ours) |
|---|---|---|---|---|
| Acc(%) | 62.81 | 64.05 | 65.22 | 64.09 |

**Effect of Different Number of Training Classes with Varying Groups:** We analyze the effect of different number of classes on training set. We utilize PudNet to predict parameters for ConvNet-3. The results are shown in Table 10. Train-C20 denotes that the training set involving 50000 groups of datasets contains 20 classes in total. Similarly, Train-C80 denotes that the training set involving 50000 groups of datasets has 80 classes. We find that with more classes included in training set, the performance of our PudNet is improved as the number of dataset groups increases.

Table 10: Effect of different number of training classes with varying groups on CIFAR100-set.

| Groups | 50 | 100 | 500 | 1000 | 5000 | 10000 | 50000 |
|---|---|---|---|---|---|---|---|
| Train-C20 | 45.77±0.33 | 49.25±0.31 | 51.31±0.28 | 51.11±0.28 | 50.93±0.29 | 51.02±0.22 | 51.10±0.19 |
| Train-C80 | 45.37±0.54 | 51.99±0.47 | 58.98±0.44 | 60.44±0.45 | 62.42±0.39 | 62.84±0.41 | 64.09±0.40 |

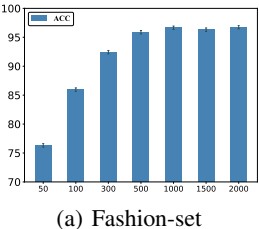 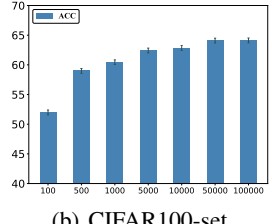 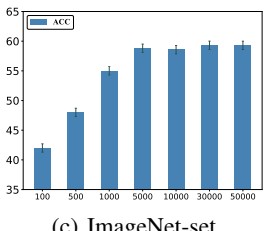

(a) Fashion-set       (b) CIFAR100-set       (c) ImageNet-set

Figure 6: Effect of Different Groups of Datasets for Training to Predict Parameters for ConvNet-3.

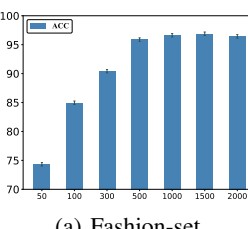 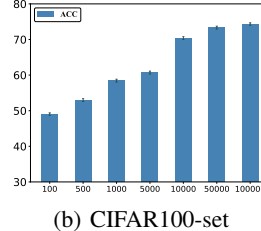 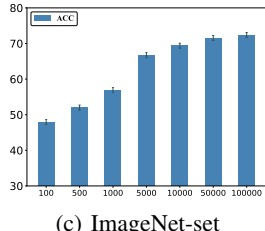

(a) Fashion-set       (b) CIFAR100-set       (c) ImageNet-set

Figure 7: Effect of Different Groups of Datasets for Training to Predict Parameters for ResNet-18.

**Effect of Different Number of Stacked GRU:** We exploit stacked GRU to transfer context-aware information of parameters for ResNet-18. We conduct experiments to investigate the effect of different number of layers in stacked GRU and report the results in Table 11. We can observe that the performance falls after rising, as the number of layers increases.

Table 11: Effect of different layers of stacked GRU.

| Stacked GRU | Target Network | num_layers=1 | num_layers=2 | num_layers=4 | num_layers=6 | num_layers=8 |
|---|---|---|---|---|---|---|
| CIFAR100-set | ResNet-18 | 68.89±0.52 | **73.33**±0.54 | 71.78±0.57 | 70.15±0.49 | 70.81±0.50 |
| ImageNet-set | ResNet-18 | 67.81±0.77 | 68.13±0.69 | **71.57**±0.71 | 70.54±0.74 | 69.26±0.71 |

**Effect of Different Structure of Feature Extractors:** To study the influence of feature extractors with different architectures, we add an experiment on CIFAR100-set in terms of ConvNet-3. The results are listed in Table 12. In Table 12, "2conv", "3conv", "4conv" denote stacking 2, 3, 4 convolution layers as the feature extractor respectively. "1linear" denotes adding one linear layer after the convolution layer. We observe that our method obtains comparable performance with the feature extractors of different structures. Thus, our method is not sensitive to the structure of feature extractors.

Table 12: Effect of different structure of feature extractors

| Method | 2conv | 3conv | 4conv | 1conv-1linear | 2conv-1linear | 3conv-1linear |
|---|---|---|---|---|---|---|
| Acc(%) | 63.87 | 63.49 | 62.44 | **64.09** | 64.01 | 63.43 |

**Effect of Different Groups of Datasets:** We analyze the effect of different groups of datasets for training. The results of utilizing PudNet to predict parameters for ConvNet-3 are shown in Figure 6. For the target network ConvNet-3, we construct 2000, 100000, 50000 groups of datasets for training PudNet on Fashion-set, CIFAR100-set, Imagenet-set respectively. Figure 7 reports the results of exploiting PudNet to predict parameters for ResNet-18. We construct 2000, 100000, 100000 groups of datasets for training our PudNet on Fashion-set, CIFAR100-set, Imagenet-set respectively. We find that with more datasets for training, our PudNet could obtain better performance. This is because

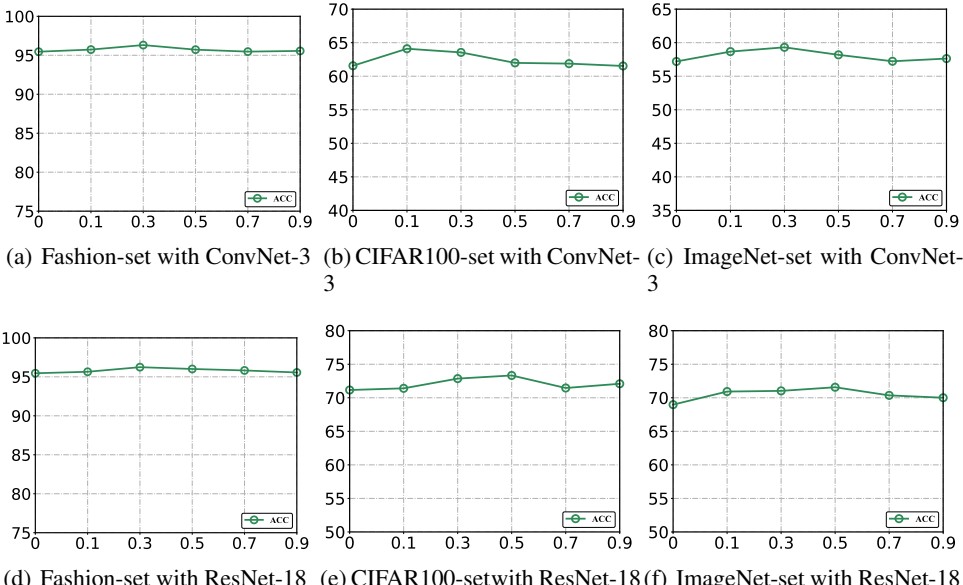

(a) Fashion-set with ConvNet-3 (b) CIFAR100-set with ConvNet-3 (c) ImageNet-set with ConvNet-3

(d) Fashion-set with ResNet-18 (e) CIFAR100-setwith ResNet-18 (f) ImageNet-set with ResNet-18

Figure 8: Sensitivity Analysis of Hyper-parameter $\eta$.

with larger groups of datasets to learn the hyper-mapping relation, our PudNet could obtain better generalization ability. However, when the number of group becomes large, the performance increase becomes slow.

**Parameter Sensitive Analysis:** We analyze the effect of different values of the hyper-parameter $\eta$. Recall that $\eta$ controls the percent of dataset complementary information in the initial residual connection. Figure 8(a)(b)(c) show the results in terms of ConvNet-3, and Figure 8(d)(e)(f) give the results for ResNet-18. We observe that our model obtain better performance when $\eta > 0$ in general. Additionally, our method is not sensitive to $\eta$ in a relatively large range.

**Convergence Analysis:** We discuss the convergence property of the proposed method by plotting the loss curves with increasing iteration. Here we utilize PudNet to predict parameters for ConvNet-3, based on Fashion-set, CIFAR100-set and ImageNet-set respectively. As shown in Figure 9, the training metric-based loss and training total loss first decrease rapidly as the number of iterations increases, and then gradually decreases to convergence.

## A.4 ALGORITHM PSEUDO-CODE

We provide the training procedure of our PudNet as listed in Algorithm 1. For each training dataset $D_i \in \mathcal{D}^{train}$, we first derive the skecth $s_i$ of dataset $D_i$ and set the initial hidden state $h_0 = s_i$ in GRU. Then, we predict the parameters of each layer in the target network $\Omega$. Finally, we optimize the the learnable parameters $\theta, \varphi$ by the the overall multi-task loss $\mathcal{L}_{total}$.

## A.5 DISCUSSION WITH SOME RELATED TOPICS

**Generalization:** The generalization ability of model is an important research topic in the machine learning community. To generalize well on unseen data, many methods have been proposed. For example, normalization methods such as batch normalization (Bjorck et al., 2018) and layer normalization (Xu et al., 2019) could improve the generalization ability (Lyu et al., 2022). Besides, some regularization techniques such as L2 regularization (Cortes et al., 2012) and dropout (Baldi & Sadowski, 2013) could also help the generalization (Wei et al., 2019). What's more, some works resort to unsupervised pretraining on large-scale data to obtain a model with great generalization ability (Devlin et al., 2018).

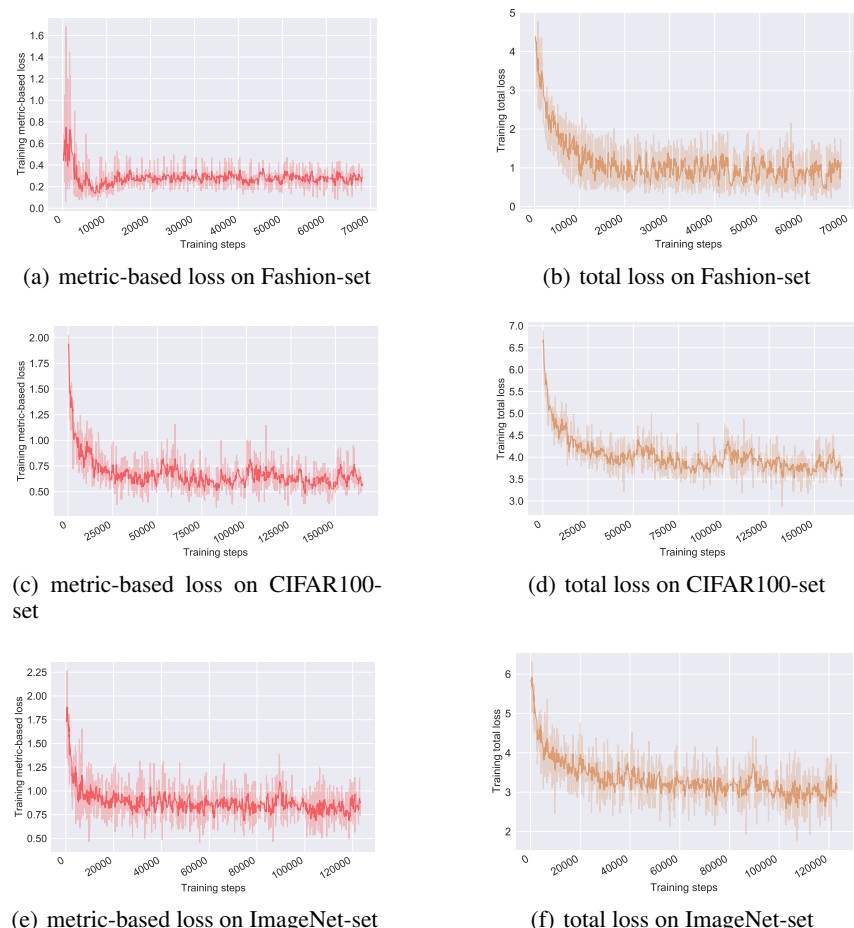

Figure 9: Training Loss of PudNet.

---

**Algorithm 1** The Training of PudNet
___
**Input:** a set of training datasets $\mathcal{D}^{train} = \{D_i\}_{i=1}^{\mathcal{N}}$, target network architecture $\Omega$.
  Initialize the learnable parameter $\theta$ of $H_\theta$.
  Initialize the learnable parameter $\varphi$ of the auxiliary full classification.
  **while** *not converged* **do**
    **for** $i \in \{1, \cdots, \mathcal{N}\}$ **do**
        Obtain a skecth $s_i$ of dataset $D_i$ via the dataset compression module;
        Initialize the hidden state $h_0 = s_i$;
        **for** *each layer $t$ in $\Omega$* **do**
            Predict the parameters $w_t$ of the $t$-th layer by (5);
            Fix the predicted parameters $w_t$ to $\Omega$;
        **for** *each batch $b$ in $D_i$* **do**
            Compute the parameter-free loss $\mathcal{L}_1$ with a batch size of $b$;
            Compute the full classification loss $\mathcal{L}_2$ and consistency loss $\mathcal{L}_3$ with a batch size of $b$;
            Update the learnable parameters $\theta, \varphi$ by the overall multi-task loss $\mathcal{L}_{total}$ in (10);
**Output:** The PudNet $H_\theta$.
___

**Transfer Learning:** The key idea of transfer learning is to transfer knowledge from source domains to a different but related target domain to improve the performance of the target learner (Zhuang et al., 2020). There are considerable methods on transfer learning, including feature-based methods (Gretton et al., 2012) , parameter-based methods (Tommasi & Caputo, 2009) and relational-based methods (Richardson & Domingos, 2006), etc. Feature-based approaches usually transform the original sample features in different domains into a common latent feature space. Parameter-based methods usually learn to finetune the parameters of the last few layers across different domain datasets. Relational-based methods transfer the logical relationship or rules learned in the source domain to the target domain.

**Meta-Learning:** Meta-Learning introduces the mechanism of "learning to learn ", which intends to train a model on a variety of learning tasks, such that it can solve new learning tasks using only a small number of training samples (Hospedales et al., 2021). Meta-Learning is usually divided into three categories: optimization-based methods, metric-based methods, and model-based methods (Yao et al., 2020). Optimization-based methods usually train the model to be easy to fine-tune by a small number of gradient steps with a small amount of training data (Finn et al., 2017). Metric-based methods learn to compare validation points with training points and predicting the label of matching training points (Hospedales et al., 2021). Note that few-shot learning can be regarded as the applications of metric-based meta-learning(Hospedales et al., 2021). Model-based methods embed the training data into activation state, making predictions for test data based on this state.

**Zero-shot Learning:** Zero-shot learning aims to learn a classifier that could classify never seen classes during training without knowing any labeled data of novel class (Wang et al., 2019). The representative zero-shot learning approaches includes: ESZSL (Romera-Paredes & Torr, 2015), SAE (Kodirov et al., 2017) ,f-CLSWGAN (Xian et al., 2018), etc. The core idea of zero-shot is to transfer the learned knowledge of seen classes to the classes unseen during training (Pourpanah et al., 2022). Since there is no label information for the unseen class, the auxiliary information for each unseen class is necessary to solve zero-shot learning problem (Wang et al., 2019). For example, given the auxiliary information for a unseen class zebra: "look like horse, with stripes", the zero-shot learning model could use this semantic information to recognize the zebra class as long as the model known the pattern of "horse" and "stripes" (Fu et al., 2015).

Since our work aims to learn a hyper-mapping between datasets and their corresponding network parameters and directly predict the parameters for an unseen dataset based on the hyper-mapping, our task is totally different from the above works. Despite this, there are some relations between our method and the above works. First, since it is prohibitive to prepare thousands of datasets and training networks on them to obtain the corresponding ground-truth parameters, we develop a new learning manner, motivated by meta-learning. Besides, because our method can directly predict network parameters for an unseen dataset, it is potential to couple with zero-shot learning, which is worthy to be further studied.

