# OpenReview forum: "Learning to Predict Parameter for Unseen Data"
_ICLR.cc/2023/Conference — Submitted to ICLR 2023_

### Official Review · Reviewer_VS2g · 2022-10-24

**Confidence:** 3
**Correctness:** 3
**Technical Novelty And Significance:** 3
**Empirical Novelty And Significance:** Not applicable
**Recommendation:** 5

**Clarity, Quality, Novelty And Reproducibility:**

Clarity:
The paper is well written and easy to follow. The reviewer has only some minor comments suggestions re clarity:

- Any reason why the authors refer to dataset embeddings as sketches and not as embeddings?

- It would be nice to present the details of the dataset used to validate the ideas in the introduction.

Quality:
The quality of the paper could be improved along two axes: (1) Improved positioning w.r.t. prior art, and (2) stronger validation.

- The core of the idea is to predict parameters for unseen datasets while training the model on seen datasets. This idea is at core of other well studied topics in ML that go beyond hypernetworks and efficient optimization. The authors should discuss how the introduced ideas relate to topics such as generalization, transfer learning, few-shot learning and zero-shot learning.

- Introduction. The paragraph starting with: “In this paper, we investigate a new training paradigm for deep neural networks” should contrast what is new in current submissions with the line of works of GHN for network parameter prediction (Ha et al., 2017; Knyazev et al., 2021).

- A relevant paper to consider could be: https://arxiv.org/pdf/1606.02185.pdf

- The paper is validated on rather simple scenarios where the training and the testing datasets are derived from a single dataset e.g. ImageNet. Only one relatively simple cross domain setup is considered (CIFAR-100 -> Animals-10). Adding additional more challenging scenarios would benefit the paper, e.g. training on ImageNet and validating on places datasets and vice-versa.

- Similarly, adding more challenging architectures would also strengthen the validation of the paper, e.g. ResNet50, ResNet101, ViT, ResNext, etc.

- The importance of dataset embedding is not ablated. Adding ablation of a variety of dataset embeddings would strengthen the paper.

- In the current dataset embedding function, how important is the choice of feature extractor? Which extractors have been considered? Is the feature extractor pretrained?

Novelty:
The paper could have some edge of novelty however the positioning of the idea should be strengthened by differentiating the current pipeline w.r.t. transfer learning, few-shot and zero-shot learning ideas.

Reproducibility:
There is no mentioning about code release. However, based on a reading of the paper, there seem to be enough details to enable the reproduction of the reported results.


**Strength And Weaknesses:**

Strengths:
The topic studied in the paper is of interest to the ML community.
The paper is well written and easy to follow

Weaknesses (details in next section):
The validation of the paper could be strengthened by considering more challenging setups and models.
Positioning the paper w.r.t to transfer learning, few-shot and zero-shot literature is missing


**Summary Of The Paper:**

The paper proposes an approach to predict parameters of the model for unseen datasets given a fixed model. To do so, the paper proposes to leverage hypernetworks to predict model parameters for unseen datasets. The proposed pipeline is validated on three sets of datasets simulated from standard ML datasets (CIFAR-100, Fashion-MNIST, and ImageNet) and three simple CNN architectures (3-layer CNN, ConvNet-3, and a Resnet18). The reported results highlight the benefits of the proposed pipeline. Moreover, the authors stress-test the pipeline on one cross domain setup by training the model on CIFAR100 and testing it om Animals 10.

**Summary Of The Review:**

Overall, the paper is interesting, well written and easy to follow. However, prior to recommending the paper for acceptance the reviewer would like to see improved validation and better positioning of the introduced idea/pipeline.

---

> ### Author Response · Authors · 2022-11-18
> **Response To Reviewer VS2g (Part 4)**
>
> > Q7: In the current dataset embedding function, how important is the choice of feature extractor? Which extractors have been considered? Is the feature extractor pretrained?
>
> A7: Thanks for your advice. For ConvNet-3, our feature extractor includes one convolutional layer and one linear layer. For ResNet-18 and ResNet-34, the feature extractor is composed of five convolutional layers and one linear layer. We train the feature extractor with our PudNet in an end-to-end fashion. More details of the feature extractors can be found in Appendix A.1.
>
> To study the influence of feature extractors with different architectures, we add an experiment on CIFAR100-set in terms of ConvNet-3. The results are listed in the following table. Here "2conv", "3conv", "4conv" denote stacking 2, 3, 4  convolution layers  as the feature extractor respectively. "1linear" denotes adding one linear layer after the convolution layer.
>
> | Method | 2conv | 3conv | 4conv | 1conv+1linear | 2conv+1linear | 3conv+1linear |
> | :----: | :---: | :---: | :---: | :-----------: | :-----------: | :-----------: |
> | Acc(%) | 63.87 | 63.49 | 62.44 |     64.09     |     64.01     |     63.43     |
>
> From the above table, we can observe that our method obtains comparable performance with the feature extractors of different structures. Thus, our method is not sensitive to the structure of feature extractors. We have added the results in Appendix A.3.
>
> > Q8: Any reason why the authors refer to dataset embeddings as sketches and not as embeddings?
>
> A8: The reason for calling the embedding of a dataset as 'sketch' stems from the work in [24].
> Following this work, we regard the embedding of a dataset as a sketch.
>
>
>
> References:
>
> [1] Understanding batch normalization (NeurIPS 2018)
>
> [2] Understanding and improving layer normalization (NeurIPS 2019)
>
> [3] Understanding the Generalization Benefit of Normalization Layers: Sharpness Reduction (NeurIPS 2022)
>
> [4] L2 regularization for learning kernels (UAI 2009)
>
> [5] Understanding dropout (NeurIPS 2013)
>
> [6] Regularization matters: Generalization and optimization of neural nets vs their induced kernel (NeurIPS 2019)
>
> [7] BERT: Pre-training of Deep Bidirectional Transformers for Language Understanding (NAACL 2019)
>
> [8] Learning transferable visual models from natural language supervision (ICML 2021)
>
> [9] A comprehensive survey on transfer learning. (Proc. IEEE 2020).
>
> [10] Optimal kernel choice for large-scale two-sample tests (NeurIPS 2012).
>
> [11] The more you know, the less you learn: From knowledge transfer to one-shot learning of object categories (BMVC 2009).
>
> [12] Markov logic networks (MLJ 2006)
>
> [13] Meta-learning in neural networks: A survey (TPAMI 2021)
>
> [14] Automated Relational Meta-learning (ICLR 2020)
>
> [15] Model-agnostic meta-learning for fast adaptation of deep networks (ICML 2017)
>
> [16] A Survey of Zero-Shot Learning: Settings, Methods, and Applications (ACM TIST 2019)
>
> [17] An embarrassingly simple approach to zero-shot learning (ICML 2015)
>
> [18] Semantic Autoencoder for Zero-Shot Learning (CVPR 2017)
>
> [19] Feature generating networks for zero-shot learning (CVPR 2018)
>
> [20] A Review of Generalized Zero-Shot Learning Methods (TPAMI 2022)
>
> [21] Zero-shot object recognition by semantic manifold distance (CVPR 2015)
>
> [22] Parameter prediction for unseen deep architectures (NeurIPS 2021)
>
> [23] Toward a neural statistician (ICLR 2017)
>
> [24] Simple and deterministic matrix sketching (KDD 2013)
>
> [25] Improved approximation algorithms for large matrices via random projections (FOCS 2006)

---

> ### Author Response · Authors · 2022-11-18
> **Response To Reviewer VS2g (Part 3)**
>
> > Q5: Similarly, adding more challenging architectures would also strengthen the validation of the paper, e.g. ResNet50, ResNet101, ViT, ResNext, etc.
>
> A5: Thanks for your advice. We add another experiment to generate parameters for a deeper architecture, ResNet-34, by PudNet on CIFAR100-set. The results are listed as:
>
> |    Method     |           |     ACC(%)     | time(sec.) |
> | :-----------: | :-------: | :------------: | :--------: |
> |  Pretrained   |           | 65.03$\pm$0.53 |     -      |
> | Meta-baseline |           | 67.40$\pm$0.69 |     -      |
> | Meta-DeepDBC  |           | 69.64$\pm$0.75 |     -      |
> |     MUSML     |           | 66.39$\pm$0.59 |    3.11    |
> |     Adam      | 1 epochs  | 47.39$\pm$1.36 |    5.47    |
> |    Scratch    | 30 epochs | 71.17$\pm$0.53 |   153.87   |
> |               | 50 epochs | 78.72$\pm$0.71 |   263.25   |
> |      GC       | 1 epochs  | 48.44$\pm$1.41 |    5.52    |
> |    Scratch    | 30 epochs | 72.37$\pm$0.75 |   154.19   |
> |               | 50 epochs | 79.85$\pm$0.83 |   264.03   |
> |    PudNet     |           | 72.87$\pm$0.64 |    0.59    |
>
> We can observe that  our method achieves comparable performance to that of GC at 30 epochs, while our method is more than 250 times faster than GC. This further demonstrates the efficiency of our method. We have added these results in Appendix A.3. Owing to the rebuttal due, we employ ResNet-34 as our target model for experiment, more deeper architectures, e.g.ResNet-50, can be studied in our future work.
>
> > Q6:  The importance of dataset embedding is not ablated. Adding ablation of a variety of dataset embeddings would strengthen the paper.
>
> A6: Thanks for your advice. We actually have conduct ablation study to study the importance of dataset embedding.
> The results is shown in Table 4 of our paper. We find that the performance of PudNet decreases when randomly initializing the hidden state of GRU. It validates that setting the dataset sketch embedding as the initial hidden state of GRU brings the performance gain.
>
> To further study the effect of dataset embedding, We design another three variants to predict parameters for ConvNet-3 on CIFAR100. "Sum" denotes summing up the representations of all samples in a dataset as the dataset embedding.  "Geometric means" denotes using the geometric mean of sample representations as the dataset embedding. "Mean+Var" denotes concatenating the mean and the variance of sample representations as the dataset embedding. The results are reported as follow:
>
> | Method |  Sum  | Geometric mean | Mean+Var | Mean(ours) |
> | :----: | :---: | :------------: | :------: | :--------: |
> | Acc(%) | 62.81 |     64.05      |  65.22   |   64.09    |
>
> From the above table, we find that these four dataset embedding methods have comparable results.
> We have added the results and analysis in Appendix A.3. Here we only explore some simple dataset embedding methods. In the future, more complicated data compression methods could be explored, such as matrix sketching [24], random projection [25]  and statistic network [23], etc.

---

> ### Author Response · Authors · 2022-11-18
> **Response To Reviewer VS2g (Part 2)**
>
> > Q2: The paragraph starting with: “In this paper, we investigate a new training paradigm for deep neural networks” should contrast what is new in current submissions with the line of works of GHN for network parameter prediction (Ha et al., 2017; Knyazev et al., 2021).
>
> A2: Thanks for your advice. We have added more discussion in Related Work of the revised paper. The original goal of hypernetwork proposed in (Ha et al., 2017) is to decrease the number of training parameters, by training a hypernetwork with a smaller size to generate
> the parameters of another network with a larger size on a fixed dataset. GHN-2 (Knyazev et al., 2021) attempts  to build a mapping between the network architectures and  network parameters, where the dataset is always fixed. GHN-2 leverages graph neural networks to model the information of the network architectures for learning the mapping. Our work is orthogonal to GHN-2, since we aim to build a mapping between the datasets and the network parameters, given a network architecture. Moreover, we extend the traditional hypernetwork by incorporating GRU to capture the relations among parameters of different layers and develop a meta-learning based manner to optimize the hypernetwork.
>
>
>
> > Q3: A relevant paper to consider could be: https://arxiv.org/pdf/1606.02185.pdf
>
> A3:Thanks for your comments. We have added this work to the page 5 of our paper.
> This paper [23] proposes a statistic network which takes a set of vectors as input and outputs the statistic information (the mean and variance of a Gaussian distribution) of this set of vectors. Different from this work, our method focuses on learning a hyper-mapping between the datasets and their corresponding network parameters, so as to directly predict network parameters for an unseen dataset. However, it will be interesting to apply the statistic network [23] to obtain the embedding of a dataset in our method. We have added this in the revised paper.
>
>
>
> > Q4: The paper is validated on rather simple scenarios where the training and the testing datasets are derived from a single dataset e.g. ImageNet. Only one relatively simple cross domain setup is considered (CIFAR-100 -> Animals-10). Adding additional more challenging scenarios would benefit the paper, e.g. training on ImageNet and validating on places datasets and vice-versa.
>
> A4: Thanks for your suggestion. Besides CIFAR-100 -> Animals-10, we actually also provided the performance of our method on the  CIFAR-100 -> CIFAR-10 dataset. Due to space limitation, we put them in Appendix A.3 of our original paper.
>
> To further validate the efficacy of our method, we  add another two experiments  on two cross-domain datasets: ImageNet -> Animals-10 and Imagenet ->CIFAR-10.  The results are as follows:
>
> |    Method    |           | ImageNet$\to$Animals10 |             | ImageNet$\to$CIFAR10 |             |
> | :----------: | :-------: | :--------------------: | :---------: | :------------------: | :---------: |
> |              |           |        ACC (%)         | time (sec.) |       ACC (%)        | time (sec.) |
> |  Pretrained  |           |     34.79$\pm$0.49     |      -      |    34.54$\pm$0.63    |      -      |
> | Meta-DeepDBC |           |     38.57$\pm$0.55     |      -      |    40.93$\pm$0.71    |      -      |
> |    MUSML     |           |     36.89$\pm$0.44     |    3.21     |    37.12$\pm$0.65    |    5.34     |
> |              | 1 epochs  |     22.09$\pm$0.58     |    31.29    |    21.37$\pm$0.71    |    60.79    |
> |     Adam     | 5 epochs  |     49.12$\pm$0.08     |   156.56    |    33.76$\pm$0.44    |   310.79    |
> |   Scratch    | 10 epochs |     66.44$\pm$0.37     |   311.74    |    48.21$\pm$0.25    |   621.75    |
> |              | 20 epochs |     73.47$\pm$0.67     |   623.92    |    65.17$\pm$0.55    |   1248.23   |
> |              | 1 epochs  |     23.01$\pm$1.02     |    30.79    |    21.44$\pm$1.02    |    60.99    |
> |      GC      | 5 epochs  |     49.77$\pm$0.54     |   155.34    |    34.41$\pm$0.78    |   310.33    |
> |   Scratch    | 10 epochs |     68.56$\pm$0.39     |   310.29    |    49.89$\pm$0.53    |   622.01    |
> |              | 20 epochs |     75.04$\pm$0.61     |   623.33    |    66.78$\pm$0.47    |   1248.98   |
> |    PudNet    |           |     42.43$\pm$0.58     |    0.48     |    45.07$\pm$0.70    |    0.57     |
>
> Based on the above table, our method still achieves surprisingly good efficiency. For instance, it takes 622.01 GPU seconds to train ResNet-18 on the ImageNet -> CIFAR10 dataset using GC from scratch and the network obtains a top-1 accuracy of
> 49.89%, while our method PudNet costs only 0.57 GPU seconds to predict the network parameters of ResNet-18 achieving comparable performance (45.07%), more than 1000 times faster than the traditional training paradigm. In addition, we could find that our method also outperforms the state-of-the-art meta-learning methods in a large margin. We have added these results to Appendix A.3.

---

> ### Author Response · Authors · 2022-11-18
> **Response To Reviewer VS2g (Part 1)**
>
> > Q1: The core of the idea is to predict parameters for unseen datasets while training the model on seen datasets. This idea is at core of other well studied topics in ML that go beyond hypernetworks and efficient optimization. The authors should discuss how the introduced ideas relate to topics such as generalization, transfer learning, few-shot learning and zero-shot learning.
>
> A1:  Thanks for your advice. Here we discuss the relation between our work and these topics. We have added this discussion to Appendix A.5.
>
> Generalization:  The generalization ability of model is an important research topic in the machine learning community. To generalize well on unseen data, many methods have been proposed. For example, normalization methods such as batch normalization [1] and layer normalization [2] could improve the generalization ability [3]. Besides, some regularization techniques such as L2 regularization [4] and dropout [5] could also help the generalization [6]. What's more, some works resort to unsupervised pretraining on large-scale data to obtain a model with great generalization ability [7,8].
>
> Transfer Learning:   The key idea of transfer learning is to transfer knowledge from source domains to a different but related target domain to improve the performance of the target learner [9]. There are considerable methods on transfer learning, including feature-based methods [10], parameter-based methods [11] and relational-based methods[12], etc. Feature-based approaches usually transform the original sample features in different domains into a common latent feature space. Parameter-based methods usually learn to finetune the parameters of the last few layers across different domain datasets. Relational-based methods transfer the logical relationship or rules learned in the source domain to the target domain.
>
> Meta-Learning: Meta-Learning introduces the mechanism of “learning to learn ", which intends to train a model on a variety of learning tasks, such that it can solve new learning tasks using only a small number of training samples[13]. Meta-Learning is usually divided into three categories: optimization-based methods, metric-based methods, and model-based methods[14]. Optimization-based methods usually train the model to be easy to fine-tune by a small number of gradient steps with a small amount of training data[15]. Metric-based methods learn to compare validation points with training points and predicting the label of matching training points[13]. Note that few-shot learning can be regarded as the applications of metric-based meta-learning[13].
> Model-based methods embed the training data into activation state, making predictions for test data based on this state[13].
>
> Zero-shot learning aims to learn a classifier that could classify never seen classes during training without knowing any labeled data of novel class [16]. The representative zero-shot learning approaches includes: ESZSL [17], SAE [18] ,f-CLSWGAN [19], etc. The core idea of zero-shot is to transfer the learned knowledge of seen classes to the classes unseen during training [20]. Since there is no label information for the unseen class, the auxiliary information for each unseen class is necessary to solve zero-shot learning problem [16]. For example, given the auxiliary information for a unseen class zebra: "look like horse, with stripes", the zero-shot learning model could use this semantic information to recognize the zebra class as long as the model has known the pattern of "horse" and "stripes" [21].
>
> Since our work aims to learn a hyper-mapping between datasets and their corresponding network parameters and directly predict the parameters for an unseen dataset based on the hyper-mapping, our task is totally different from the above works. Despite this, there are some relations between our method and the above works. First,  since it is prohibitive to prepare thousands of datasets and training networks on them to obtain the corresponding ground-truth parameters, we develop a new learning manner, motivated by meta-learning.
> Besides, because our method can directly predict network parameters for an unseen dataset, it is potential to couple with zero-shot learning, which is worthy to be further studied.

---

> ### Author Response · Authors · 2022-11-22
> **Discussion Reminder**
>
> We sincerely thank you for your efforts to review our paper. We gently remind the reviewer that we tried our best to address your concerns via our responses and the revision of the paper. We would be most delighted to hear more from you if there are any further concerns.

---

### Official Review · Reviewer_AS8e · 2022-10-25

**Confidence:** 3
**Correctness:** 2
**Technical Novelty And Significance:** 3
**Empirical Novelty And Significance:** 2
**Recommendation:** 5

**Clarity, Quality, Novelty And Reproducibility:**

Regarding writing, the clarity and quality is good. Hypernetwork supporting multiple datasets is novel.

**Strength And Weaknesses:**

- Strengths
    - Writing is clear and easy to follow.
    - Novel concept idea. This work reveals that there is correlation between datasets and the network parameters.
    - With a single hypernetwork, this work can predict the parameters of multiple datasets. I think this is quite new.
- Weaknesses
    - The main concern is that the experiments to validate the proposed method are limited and weak.
        - While experimental setting is similar with the existing meta-learning methods for the classification task, the most recent or powerful meta-learning methods are not compared.
        - In addition, I think that such similarity with the existing few-shot classificaiton tasks for the meta-learning are not attractive and not challenging. The most architectures are small (3 CNN layer ~ ResNet-18) and the small number of classes in each task, most unseen tasks are sampled from the in-distribution dataset with the seen tasks. I think this work need to more focus on the cross-domain setting or different architectures.
        - For the fair comparison, I think Meta-dataset [1] can be used.
     - There is a related work [2] that a single hypernetwork can predict parameters for unseen architectures. I recommend to add discussion with this related work.

[1] Meta-Dataset: A Dataset of Datasets for Learning to Learn from Few Examples, ICLR 2020
[2] Parameter Prediction for Unseen Deep Architectures, NeurIPS 2021

**Summary Of The Paper:**

To overcome heavy burden of computational cost and time to learn network parameters, this work propose a method that predicts parameters of the network on the given unseen dataset. For this, they allow the new hypernetwork to directly predict the network parameters with forward process, by learning mapping between datasets and their corresponding network parameters. With only 0.5 GPU seconds, this method can predict the network parameters of ResNet-18 which is competitive performance compared with the parameters trained on the dataset from scratch.

**Summary Of The Review:**

I think the experimental setting is rather limited to support or validate the proposed method. As I mentioned in the strengths and weaknesses section, I hope the authors address my concern by showing the solid and extensive experimental results on more realistic settings.

---

> ### Author Response · Authors · 2022-11-18
> **Response To Reviewer AS8e (Part 2)**
>
> > Q2: In addition, I think that such similarity with the existing few-shot classificaiton tasks for the meta-learning are not attractive and not challenging. The most architectures are small (3 CNN layer ~ ResNet-18) and the small number of classes in each task, most unseen tasks are sampled from the in-distribution dataset with the seen tasks. I think this work need to more focus on the cross-domain setting or different architectures.
>
> Moreover, we add another experiments to generate parameters for a deeper architecture, ResNet-34, by PudNet on CIFAR100-set. The results are listed as:
>
> |    Method     |           |     ACC(%)     | time(sec.) |
> | :-----------: | :-------: | :------------: | :--------: |
> |  Pretrained   |           | 65.03$\pm$0.53 |     -      |
> | Meta-baseline |           | 67.40$\pm$0.69 |     -      |
> | Meta-DeepDBC  |           | 69.64$\pm$0.75 |     -      |
> |     MUSML     |           | 66.39$\pm$0.59 |    3.11    |
> |     Adam      | 1 epochs  | 47.39$\pm$1.36 |    5.47    |
> |    Scratch    | 30 epochs | 71.17$\pm$0.53 |   153.87   |
> |               | 50 epochs | 78.72$\pm$0.71 |   263.25   |
> |      GC       | 1 epochs  | 48.44$\pm$1.41 |    5.52    |
> |    Scratch    | 30 epochs | 72.37$\pm$0.75 |   154.19   |
> |               | 50 epochs | 79.85$\pm$0.83 |   264.03   |
> |    PudNet     |           | 72.87$\pm$0.64 |    0.59    |
>
> We can observe that  our method achieves comparable performance to that of GC at 30 epochs, while our method is more than 250 times faster than GC. This further demonstrates the efficiency of our method. We have added these results in Appendix A.3.
>
> > Q3: For the fair comparison, I think Meta-dataset can be used.
>
> A3:Thanks for your advice.  To further validate the efficacy of our method, we add another two experiments on two cross-domain datasets: ImageNet -> Animals-10 and Imagenet->CIFAR-10, as shown in the above response. The experimental results once again demonstrate the efficacy and efficiency of our method.
>
> We studied the meta-dataset and found that it would take a lot of time to process this dataset. Due to the limited time in rebuttal phase, we would like to further evaluate our method on this dataset in our future work.
>
> > Q4: There is a related work [2] that a single hypernetwork can predict parameters for unseen architectures. I recommend to add discussion with this related work.
>
> A4: Thanks for your advice. GHN-2 [2] attempts  to build a mapping between the network architectures and  network parameters, where the dataset is always fixed. GHN-2 leverages graph neural networks to model the information of the network architectures for learning the mapping. Our work is orthogonal to GHN-2, since we aim to build a mapping between the datasets and the network parameters, given a network architecture. Moreover, we extend the traditional hypernetwork by incorporating GRU to capture the relations among parameters of different layers and develop a meta-learning based manner to optimize the hypernetwork. We add this in Page 3 of the revised paper.
>
>
>
> References:
>
> [1] Subspace Learning for Effective Meta-Learning (ICML 2022)
>
> [2] Parameter prediction for unseen deep architectures (NeurIPS 2021)

---

> > ### Comment · Reviewer_AS8e · 2022-11-27
> > **Thank the authors for the response.**
> >
> > The authors addressed my concerns partially by providing the comparison with the recent meta-learning methods and discussion about the paper [2]. However, I think ImageNet 1K --> Animals10 and ImageNet --> CIFAR10 are not truely cross-domain experiments as ImageNet includes various animal classes and the property of ImageNet 1K and CIFAR10 is similar as coarse classification datasets.
> > I hope the authors address my remaining concern.

---

> > > ### Author Response · Authors · 2022-11-28
> > > **Response To Reviewer AS8e**
> > >
> > > Thanks for your further discussion. We are pleased to see that some of your concerns have been addressed.
> > >
> > > For the  cross-domain experiments, we cordially argue that the used two datasets (i.e., ImageNet -> Animals-10 and ImageNet -> CIFAR10) are cross-domain.  For the ImageNet and Animals-10 datasets, the reference [1] uses them for out-of-distribution (OOD) learning, where  Animals-10  is used to train the model, and ImageNet is regarded as an out-of-distribution dataset for testing. In the meantime, the ImageNet and CIFAR10 datasets are widely used for out-of-distribution (OOD) detection [2, 3] and cross-domain learning [4]. Moreover, CIFAR10 is a coarse classification dataset, while ImageNet contains fine-grained classes, e.g., "Newfoundland dog", "French bulldog",  "Alaskan malamute", "African hunting dog", etc. Motivated by these works, we also take them as the cross-domain datasets to evaluate our method.
> > >
> > > To further demonstrate the efficacy of our method, we add another two cross-domain experiments on the ImageNet -> DTD and CIFAR100 -> DTD datasets, respectively. DTD   is a texture classification dataset [5] to classify images into different texture categories, which has been integrated into the Meta-Dataset [6]. The label space of DTD is significantly different from that of ImageNet and CIFAR100. For DTD, we randomly select 2/3 samples from each category to directly generate the network parameters by PudNet trained on ImageNet or CIFAR100, and use the rest for testing the performance of the predicted parameters. The experimental results are listed as follows:
> > >
> > > |    Method    |           | ImageNet$\to$DTD |             | CIFAR100$\to$DTD |             |
> > > | :----------: | :-------: | :--------------: | :---------: | :--------------: | :---------: |
> > > |              |           |     ACC (\%)     | time (sec.) |     ACC (\%)     | time (sec.) |
> > > |  Pretrained  |           |  31.05$\pm$0.51  |      -      |  30.27$\pm$0.58  |      -      |
> > > | Meta-DeepDBC |           |  34.01$\pm$0.69  |      -      |  35.5$\pm$0.71   |      -      |
> > > |    MUSML     |           |  33.67$\pm$0.63  |    2.18     |  33.34$\pm$0.68  |    2.18     |
> > > |              | 5 epochs  |  24.17$\pm$0.72  |    26.01    |  24.17$\pm$0.72  |    26.01    |
> > > |     Adam     | 10 epochs |  33.26$\pm$0.90  |    51.79    |  33.26$\pm$0.90  |    51.79    |
> > > |   Scratch    | 20 epochs |  42.51$\pm$0.44  |   102.11    |  42.51$\pm$0.44  |   102.11    |
> > > |              | 50 epochs |  52.16$\pm$0.56  |   257.91    |  52.16$\pm$0.56  |   257.91    |
> > > |              | 5 epochs  |  24.43$\pm$0.83  |    25.87    |  24.43$\pm$0.83  |    25.87    |
> > > |      GC      | 10 epochs |  35.49$\pm$0.61  |    51.70    |  35.49$\pm$0.61  |    51.70    |
> > > |   Scratch    | 20 epochs |  44.05$\pm$0.52  |   102.63    |  44.05$\pm$0.52  |   102.63    |
> > > |              | 50 epochs |  53.58$\pm$0.61  |   257.24    |  53.58$\pm$0.61  |   257.24    |
> > > |    PudNet    |           |  47.50$\pm$0.71  |    0.39     |  38.05$\pm$0.73  |    0.39     |
> > >
> > > Based on the above table, we could find that although images in DTD are significantly different from those in ImageNet or CIFAR100, our method still achieves surprisingly good efficacy. For instance, it takes 257.24 GPU seconds to train ResNet-18 on the ImageNet-> DTD dataset with 50 epochs using GC from scratch, obtaining a top-1 accuracy of 53.58%. However, our PudNet costs only 0.39 GPU seconds to predict the network parameters of ResNet-18, achieving comparable performance (47.50%).  Our method is more than 600 times faster than the traditional training paradigm. We will add these results in the final version of our manuscript.
> > >
> > > References:
> > >
> > > [1] Trash to Treasure: Harvesting OOD Data with Cross-Modal Matching for Open-Set Semi-Supervised Learning (ICCV 2021)
> > >
> > > [2] Single Layer Predictive Normalized Maximum Likelihood for Out-of-Distribution Detection (NeurIPS 2021)
> > >
> > > [3] Predictive Uncertainty Estimation via Prior Networks (NeurIPS 2018)
> > >
> > > [4] Cross-domain Few-shot Learning with Task-specific Adapters (CVPR 2022)
> > >
> > > [5] Describing textures in the wild (CVPR 2014)
> > >
> > > [6] Meta-dataset: A dataset of datasets for learning to learn from few examples (ICLR 2020)

---

> ### Author Response · Authors · 2022-11-18
> **Response To Reviewer AS8e (Part 1)**
>
> > Q1: While experimental setting is similar with the existing meta-learning methods for the classification task, the most recent or powerful meta-learning methods are not compared.
>
> A1: Thanks for your comments. In our experiment, we actually compare our method with two representative  meta-learning methods, i.e., MatchNet and ProtoNet, which are widely used as the baselines. Moreover, we also compare with Meta-Baseline (ICCV 2021) and Meta-DeepDBC (CVPR 2022) which are proposed recently.
>
> To further verify the effectiveness of our method, we add a comparison with MUSML [1] (ICML 2022), which is proposed more recently. The experimental results are listed as:
>
> | Target Network | Method |  Fashion-set   |  CIFAR100-set  |  ImageNet-set  |
> | :------------: | :----: | :------------: | :------------: | :------------: |
> |   ConvNet-3    | MUSML  | 96.05$\pm$0.32 | 56.49$\pm$0.56 | 54.03$\pm$0.94 |
> |                | PudNet | 96.64$\pm$0.34 | 64.09$\pm$0.40 | 59.31$\pm$0.64 |
> |   ResNet-18    | MUSML  | 95.87$\pm$0.44 | 66.47$\pm$0.63 | 66.03$\pm$0.91 |
> |                | PudNet | 96.24$\pm$0.39 | 73.33$\pm$0.54 | 71.57$\pm$0.71 |
>
> From the above table, we can find that our method is still superior to MUSML. This shows the effectiveness of our method once again. We have added the results of MUSML in the revised paper.
>
>
>
> > Q2: In addition, I think that such similarity with the existing few-shot classificaiton tasks for the meta-learning are not attractive and not challenging. The most architectures are small (3 CNN layer ~ ResNet-18) and the small number of classes in each task, most unseen tasks are sampled from the in-distribution dataset with the seen tasks. I think this work need to more focus on the cross-domain setting or different architectures.
>
> A2:  Thanks for your suggestion. Besides CIFAR-100 -> Animals-10, we actually also provided the performance of our method on the  CIFAR-100 -> CIFAR-10 dataset. Due to space limitation, we put them in Appendix A.3 of our original paper.
>
> To further validate the efficacy of our method, we  add another two experiments  on two cross-domain datasets: ImageNet -> Animals-10 and Imagenet -> CIFAR-10. The results are reported as follows:
>
> |    Method    |           | ImageNet$\to$Animals10 |             | ImageNet$\to$CIFAR10 |             |
> | :----------: | :-------: | :--------------------: | :---------: | :------------------: | :---------: |
> |              |           |        ACC (%)         | time (sec.) |       ACC (%)        | time (sec.) |
> |  Pretrained  |           |     34.79$\pm$0.49     |      -      |    34.54$\pm$0.63    |      -      |
> | Meta-DeepDBC |           |     38.57$\pm$0.55     |      -      |    40.93$\pm$0.71    |      -      |
> |    MUSML     |           |     36.89$\pm$0.44     |    3.21     |    37.12$\pm$0.65    |    5.34     |
> |              | 1 epochs  |     22.09$\pm$0.58     |    31.29    |    21.37$\pm$0.71    |    60.79    |
> |     Adam     | 5 epochs  |     49.12$\pm$0.08     |   156.56    |    33.76$\pm$0.44    |   310.79    |
> |   Scratch    | 10 epochs |     66.44$\pm$0.37     |   311.74    |    48.21$\pm$0.25    |   621.75    |
> |              | 20 epochs |     73.47$\pm$0.67     |   623.92    |    65.17$\pm$0.55    |   1248.23   |
> |              | 1 epochs  |     23.01$\pm$1.02     |    30.79    |    21.44$\pm$1.02    |    60.99    |
> |      GC      | 5 epochs  |     49.77$\pm$0.54     |   155.34    |    34.41$\pm$0.78    |   310.33    |
> |   Scratch    | 10 epochs |     68.56$\pm$0.39     |   310.29    |    49.89$\pm$0.53    |   622.01    |
> |              | 20 epochs |     75.04$\pm$0.61     |   623.33    |    66.78$\pm$0.47    |   1248.98   |
> |    PudNet    |           |     42.43$\pm$0.58     |    0.48     |    45.07$\pm$0.70    |    0.57     |
>
> Based on the above table, our method still achieves surprisingly good efficiency. For instance, it takes 622.01 GPU seconds to train ResNet-18 on the ImageNet -> CIFAR10 dataset using GC from scratch and the network obtains a top-1 accuracy of 49.89%, while our method PudNet costs only 0.57 GPU seconds to predict the network parameters of ResNet-18 achieving comparable performance (45.07%), more than 1000 times faster than the traditional training paradigm. In addition, we could find that our method still outperforms the meta-learning methods on the cross-domain dataset. We have added these results to Appendix A.3.

---

> ### Author Response · Authors · 2022-11-22
> **Discussion Reminder**
>
> We sincerely thank you for your efforts to review our paper. We gently remind the reviewer that we tried our best to address your concerns via our responses and the revision of the paper. We would be most delighted to hear more from you if there are any further concerns.

---

### Official Review · Reviewer_y6gY · 2022-10-30

**Confidence:** 3
**Correctness:** 3
**Technical Novelty And Significance:** 3
**Empirical Novelty And Significance:** 3
**Recommendation:** 6

**Clarity, Quality, Novelty And Reproducibility:**

### Clarity
The paper is well-written and clear.

### Novelty
Although the employed techniques like the dataset compression are not new, the whole framework is novel to me.

### Reproducibility
The dataset split, training details are provided, but I don't see any code.


**Strength And Weaknesses:**

### Strengths
- I'm not familiar with the area of generating network weight for unseen datasets. But to me, this paper is well-motivated and novel.
- The organization of this paper is pretty good, first using an experiment to demonstrate the correlation between the dataset and the trained weight, then three main components of the method, it's easy to follow and interesting to read.
- Empirical evaluation is thorough and have some interesting results. Although the performance still lay far behind a fully-trained network, it can achieve reasonable results.

### Weaknesses
- The core of the proposed approach is that it can generate parameters for unseen datasets. The paper only shows CIFAR-100 -> Animals-10, I would expect some other evidence. For example, CIFAR-10 -> ImageNet.
- One interesting I notice in the method is that it trains a weight generator, which allows generating parameters of different sizes. However, I don't see any evidence in the experiment part. For instance, can you use the network to generate a larger ResNet-50 and show the accuracy?
- The ablation study reveals the effectiveness of different parts. Despite this, it could be better if the authors can provide more insights and intuitions in the method part. Some part is a bit ad-hoc to me. For instance, adding an additional classification head and trying to match the two predictions.


**Summary Of The Paper:**

This paper proposes to predict the parameter of a neural network via a hyper-network.
It mainly consists of two components, a compression method to capture the feature of a dataset, and a GRU-based network to generate the per-layer parameter.
Some empirical results are very interesting and surprising.

**Summary Of The Review:**

The paper is novel and interesting to me in general. However, there are some missing empirical evaluations and intuitions, which I would like to hear from the authors in the feedback phases.

---

> ### Author Response · Authors · 2022-11-18
> **Response To Reviewer y6gY (Part 2)**
>
> > Q2: One interesting I notice in the method is that it trains a weight generator, which allows generating parameters of different sizes. However, I don’t see any evidence in the experiment part. For instance, can you use the network to generate a larger ResNet-50 and show the accuracy?
>
> A2: Sorry for confusing you. We generate parameters of different sizes via changing the output dimensions of the weight generator. In the experiments, we actually generate parameters for different architectures including ConvNet-3 and ResNet-18. The structure of the weight generator can be found in Appendix A.1.
>
> Moreover, we add another experiments to generate parameters for a deeper architecture, ResNet-34, by PudNet on CIFAR100-set. The results are listed as:
>
> |    Method     |           |     ACC(%)     | time(sec.) |
> | :-----------: | :-------: | :------------: | :--------: |
> |  Pretrained   |           | 65.03$\pm$0.53 |     -      |
> | Meta-baseline |           | 67.40$\pm$0.69 |     -      |
> | Meta-DeepDBC  |           | 69.64$\pm$0.75 |     -      |
> |     MUSML     |           | 66.39$\pm$0.59 |    3.11    |
> |     Adam      | 1 epochs  | 47.39$\pm$1.36 |    5.47    |
> |    Scratch    | 30 epochs | 71.17$\pm$0.53 |   153.87   |
> |               | 50 epochs | 78.72$\pm$0.71 |   263.25   |
> |      GC       | 1 epochs  | 48.44$\pm$1.41 |    5.52    |
> |    Scratch    | 30 epochs | 72.37$\pm$0.75 |   154.19   |
> |               | 50 epochs | 79.85$\pm$0.83 |   264.03   |
> |    PudNet     |           | 72.87$\pm$0.64 |    0.59    |
>
> We can observe that  our method achieves comparable performance to that of GC at 30 epochs, while our method is more than 250 times faster than GC. This further demonstrates the efficiency of our method. We have added these results in Appendix A.3.
>
> Owing to the rebuttal due, we employ ResNet-34 as our target model for experiment. We would like to test our method on more deeper architectures, e.g. ResNet-50 in our future work.
>
> > Q3: The ablation study reveals the effectiveness of different parts. Despite this, it could be better if the authors can provide more insights and intuitions in the method part. Some part is a bit ad-hoc to me. For instance, adding an additional classification head and trying to match the two predictions.
>
> A3: Thanks for your advice. The idea of adding an additional classification head is inspired by TADAM [1].  Our parameter prediction task co-trained with a full classification head is related to curriculum learning [2]. Since learning on a varying label space is more challenging than learning on a static one, the full classification problem that maps features to a static label set could be regraded as a simpler curriculum. This easier 'prerequisite' could help the hypernetwork to obtain the basic level knowledge before handling harder parameter prediction task. Besides, we match the two predictions to make the predictions of our main task and the auxiliary task consistent, which is motivated by [3,4]. We have added more descriptions in Page 6 of the revised paper.
>
> References:
>
> [1] TADAM: Task dependent adaptive metric for improved few-shot learning (NeurIPS 2018)
>
> [2] Meta-learning with memory-augmented neural networks (ICML 2016)
>
> [3] ASM2TV: An Adaptive Semi-supervised Multi-Task Multi-View Learning Framework for Human Activity Recognition (AAAI 2022)
>
> [4] Large Scale Incremental Learning (CVPR 2019)

---

> ### Author Response · Authors · 2022-11-18
> **Response To Reviewer y6gY (Part 1)**
>
> > Q1:  The core of the proposed approach is that it can generate parameters for unseen datasets. The paper only shows CIFAR-100 -> Animals-10, I would expect some other evidence. For example, CIFAR-10 -> ImageNet.
>
> A1: Thanks for your suggestion. Besides CIFAR-100 -> Animals-10, we actually also provided the performance of our method on the  CIFAR-100 -> CIFAR-10 dataset. Due to space limitation, we put them in Appendix A.3 of our original paper.
> To further validate the efficacy of our method, we  add another two experiments  on two cross-domain datasets: ImageNet -> Animals-10 and Imagenet->CIFAR-10.  The results are as follows:
>
> |    Method    |           | ImageNet$\to$Animals10 |             | ImageNet$\to$CIFAR10 |             |
> | :----------: | :-------: | :--------------------: | :---------: | :------------------: | :---------: |
> |              |           |        ACC (%)         | time (sec.) |       ACC (%)        | time (sec.) |
> |  Pretrained  |           |     34.79$\pm$0.49     |      -      |    34.54$\pm$0.63    |      -      |
> | Meta-DeepDBC |           |     38.57$\pm$0.55     |      -      |    40.93$\pm$0.71    |      -      |
> |    MUSML     |           |     36.89$\pm$0.44     |    3.21     |    37.12$\pm$0.65    |    5.34     |
> |              | 1 epochs  |     22.09$\pm$0.58     |    31.29    |    21.37$\pm$0.71    |    60.79    |
> |     Adam     | 5 epochs  |     49.12$\pm$0.08     |   156.56    |    33.76$\pm$0.44    |   310.79    |
> |   Scratch    | 10 epochs |     66.44$\pm$0.37     |   311.74    |    48.21$\pm$0.25    |   621.75    |
> |              | 20 epochs |     73.47$\pm$0.67     |   623.92    |    65.17$\pm$0.55    |   1248.23   |
> |              | 1 epochs  |     23.01$\pm$1.02     |    30.79    |    21.44$\pm$1.02    |    60.99    |
> |      GC      | 5 epochs  |     49.77$\pm$0.54     |   155.34    |    34.41$\pm$0.78    |   310.33    |
> |   Scratch    | 10 epochs |     68.56$\pm$0.39     |   310.29    |    49.89$\pm$0.53    |   622.01    |
> |              | 20 epochs |     75.04$\pm$0.61     |   623.33    |    66.78$\pm$0.47    |   1248.98   |
> |    PudNet    |           |     42.43$\pm$0.58     |    0.48     |    45.07$\pm$0.70    |    0.57     |
>
> Based on the above table, our method still achieves surprisingly good efficiency. For instance, it takes 622.01 GPU seconds to train ResNet-18 on the ImageNet -> CIFAR10 dataset using GC from scratch and obtain a top-1 accuracy of 49.89%, while our method PudNet costs only 0.57 GPU seconds to predict the network parameters of ResNet-18 achieving comparable performance (45.07%), more than 1000 times faster than the traditional training paradigm. In addition, we could find that our method also outperforms the state-of-the-art meta-learning methods in a large margin. We have added these results to Appendix A.3.

---

> ### Author Response · Authors · 2022-11-22
> **Discussion Reminder**
>
> We sincerely thank you for your efforts to review our paper. We gently remind the reviewer that we tried our best to address your concerns via our responses and the revision of the paper. We would be most delighted to hear more from you if there are any further concerns.

---

### Author Response · Authors · 2022-11-18
**General Response**

We would like to thank all reviewers for their insightful and helpful feedback. We are excited that all the three reviewers thought our paper to be: "novel", "interesting" and "well written".

We have updated the first version of our revised paper based on the comments from reviewers. We sincerely hope that our work could inspire more future works in the community. Once again, we thank all the reviewers for providing constructive suggestions.

---

### Decision · Program_Chairs · 2023-01-20

**Decision:**

Reject

**Justification For Why Not Higher Score:**

n/a

**Justification For Why Not Lower Score:**

n/a

**Metareview: Summary, Strengths And Weaknesses:**

The paper proposes a new training paradigm to learn a hyper-mapping between the datasets and the optimal neural network parameters. The reviewers agree that the method is novel and interesting. However, the main concern is the validation of the method and the experiment setups. The reviewers think it is difficult to validate the improvement from the method based on the results provided in the paper. Therefore, the paper would need to be futher improved before it can be accepted to ICLR.